# Integrative analysis of cell state changes in lung fibrosis with peripheral protein biomarkers

Christoph H Mayr[1,†], Lukas M Simon[2,†], Gabriela Leuschner[1,3], Meshal Ansari[1,2], Janine Schniering[1,4], Philipp E Geyer[5], Ilias Angelidis[1] , Maximilian Strunz[1], Pawandeep Singh[1], Nikolaus Kneidinger[3], Frank Reichenberger[6], Edith Silbernagel[6], Stephan Böhm[7], Heiko Adler[8], Michael Lindner[6,9], Britta Maurer[4], Anne Hilgendorff[10], Antje Prasse[11], Jürgen Behr[3,6], Matthias Mann[5] , Oliver Eickelberg[12], Fabian J Theis[2,*]  & Herbert B Schiller[1,**]

## Abstract

The correspondence of cell state changes in diseased organs to peripheral protein signatures is currently unknown. Here, we generated and integrated single-cell transcriptomic and proteomic data from multiple large pulmonary fibrosis patient cohorts. Integration of 233,638 single-cell transcriptomes ($n = 61$) across three independent cohorts enabled us to derive shifts in cell type proportions and a robust core set of genes altered in lung fibrosis for 45 cell types. Mass spectrometry analysis of lung lavage fluid ($n = 124$) and plasma ($n = 141$) proteomes identified distinct protein signatures correlated with diagnosis, lung function, and injury status. A novel SSTR2+ pericyte state correlated with disease severity and was reflected in lavage fluid by increased levels of the complement regulatory factor CFHR1. We further discovered CRTAC1 as a biomarker of alveolar type-2 epithelial cell health status in lavage fluid and plasma. Using cross-modal analysis and machine learning, we identified the cellular source of biomarkers and demonstrated that information transfer between modalities correctly predicts disease status, suggesting feasibility of clinical cell state monitoring through longitudinal sampling of body fluid proteomes.

**Keywords** biomarker; data integration; fibrosis; proteomics; single-cell RNA-seq

**Subject Categories** Biomarkers; Proteomics; Respiratory System

## Introduction

The accumulation and persistence of scar tissue in fibrotic diseases such as pulmonary fibrosis, liver cirrhosis, and cardiovascular disease is among the most severe clinical issues, causing an estimated 45% of all deaths in the developed world (Cox & Erler, 2011). Interstitial lung diseases (ILDs) are a heterogeneous group of diseases ultimately leading to pulmonary fibrosis, which can cause severe destruction of the lung parenchyma and respiratory failure. Several potential risk factors have been identified, including genetic predisposition (Allen *et al*, 2019), smoking (Baumgartner *et al*, 1997), infections (e.g., viruses) (Sheng *et al*, 2019), aging (Selman *et al*, 2016), and autoimmunity (Fischer & du Bois, 2012; Schiller *et al*, 2017). Addressing the heterogeneity of ILD entities, disease progression and prognosis, and the currently unpredictable occurrence of acute exacerbations of disease, require new molecular approaches for personalized patient monitoring.

The recent surge of innovation in single-cell genomics enables an entirely novel cell type-specific viewpoint on pathological changes

1 Institute of Lung Biology and Disease and Comprehensive Pneumology Center with the CPC–M bioArchive, Helmholtz Zentrum München, Member of the German Center for Lung Research (DZL), Munich, Germany
2 Institute of Computational Biology, Helmholtz Zentrum München, Munich, Germany
3 Department of Internal Medicine V, Ludwig-Maximilians University (LMU) Munich, Member of the German Center for Lung Research (DZL), CPC-M bioArchive, Munich, Germany
4 Department of Rheumatology, Center of Experimental Rheumatology, University & University Hospital Zurich, Zurich, Switzerland
5 Department of Proteomics and Signal Transduction, Max Planck Institute of Biochemistry, Martinsried, Germany
6 Asklepios Fachkliniken Munich-Gauting, CPC-M bioArchive, Member of the German Center for Lung Research (DZL), Munich, Germany
7 Faculty of Medicine, Max von Pettenkofer-Institute, Virology, National Reference Center for Retroviruses, LMU München, Munich, Germany
8 Helmholtz Zentrum München, Research Unit Lung Repair and Regeneration, Member of the German Center for Lung Research (DZL), Munich, Germany
9 University Department of Visceral and Thoracic Surgery Salzburg, Paracelsus Medical University, Salzburg, Austria
10 Center for Comprehensive Developmental Care (CDeCLMU), Member of the German Center for Lung Research (DZL), Hospital of the Ludwig-Maximilians University (LMU), CPC-M bioArchive, Munich, Germany
11 Department of Pneumology, Hannover Medical School, Member of the German Center for Lung Research (DZL), Hannover, Germany
12 Division of Pulmonary, Allergy, and Critical Care Medicine, Department of Medicine, University of Pittsburgh, Pittsburgh, PA, USA
*Corresponding author. Tel: +49 89 3187 43260; E-mail: fabian.theis@helmholtz-muenchen.de
**Corresponding author. Tel: +49 89 3187 1194; E-mail: herbert.schiller@helmholtz-muenchen.de
†These authors contributed equally to this work

in disease. Based on these new technologies, the Human Cell Atlas project aims at building a comprehensive reference map of all human cells as a basis for understanding fundamental human biological processes and diagnosing, monitoring, and treating disease. This includes recent international efforts toward building a human Lung Cell Atlas in health and disease (Schiller *et al*, 2019). A first draft of the cellular composition of mouse and human lung has been established (Han *et al*, 2018; Tabula Muris Consortium *et al*, 2018; Vieira Braga *et al*, 2019; Angelidis *et al*, 2019; Travaglini *et al*, 2020), and several recent single-cell profiling studies reported cellular and molecular changes associated with pulmonary fibrosis (Reyfman *et al*, 2019; Morse *et al*, 2019). However, this nascent draft of a human Lung Cell Atlas currently lacks extension into the complexity of the proteome layer and integrated analysis of gene expression differences across large numbers of patients.

As disease trajectories in ILD patients are often highly variable, protein signatures in patient body fluids promise improved personalized treatment and longitudinal monitoring of patients (Maher *et al*, 2017; Neighbors *et al*, 2018). The transcriptomic and proteomic changes in end-stage ILD patient lung tissue have been resolved using microarrays, RNA-sequencing, and mass spectrometry (Zuo *et al*, 2002; Schiller *et al*, 2017; McDonough *et al*, 2019). Furthermore, first gene and protein expression signatures in ILD bronchoalveolar lavage (BAL), which is obtained during bronchoscopy, have been analyzed (Foster *et al*, 2015; Prasse *et al*, 2019). Currently, it is unclear which cellular and molecular processes in the lung correspond to these biomarker signatures, representing a tissue or fluid average which does not resolve cellular composition and disease-specific cell states.

In this work, we explore the idea that protein signatures found in bronchoalveolar lavage and plasma, both of which are accessible for longitudinal monitoring of patients, can be used to predict pathological cell state changes in the lung. We aimed at establishing a first proof of concept for this approach, which has relevance for future predictive and interceptive medicine (Rajewsky *et al*, 2020). Our analysis dissects human lung fibrosis at the single-cell level, defining robust differential gene expression profiles and cell frequency changes across multiple studies for ILD. Using mass spectrometry, we discover protein biomarker signatures associated with diagnosis, lung function, and injury status and predict the cellular sources of these protein signatures based on single-cell analysis. Using machine learning, we show that fluid proteome signatures are predictive of specific cell state changes in the lung.

# Results

### An integrated single-cell atlas of human lung fibrosis

To analyze transcriptional changes in lung fibrosis at cellular resolution, we obtained whole lung parenchyma single-cell suspensions using end-stage lung fibrosis tissues from three ILD patients (IPF $n = 2$, EAA $n = 1$) and non-fibrotic control tissues from 11 non-lung disease patients for comparison (further referred to as controls). Dimension reduction was used to visualize a data manifold representing the gene expression space of 41,888 single-cells, generated by using the Drop-seq workflow (Macosko *et al*, 2015; Fig 1A and B; control, $n = 11$; ILD, $n = 3$) (Appendix Fig S1A and B). We

generated subsets of the whole lung parenchyma datasets for COL1A2+ stromal cells (Fig 1C and Appendix Fig S1C–E), EPCAM+ epithelial cells (Fig 1D and Appendix Fig S1F–H), CLDN5+ endothelial cells (Fig 1E and Appendix Fig S1I–K), and CD45+ leukocytes (Fig 1F and Appendix Fig S1L–N). From these subsets, we derived cluster identities (Appendix Fig S1; Dataset EV1) that were manually annotated using previously established single-cell signatures in the human lung (Vieira Braga *et al*, 2019; Travaglini *et al*, 2020). The final annotation revealed 45 cell type identities, characterized by unique marker gene expression profiles (Fig 1G–J) that were to some extent preserved in end-stage fibrosis.

To increase statistical power, and ensure generalizability and reproducibility of our results, we integrated our dataset with two large publicly available single-cell RNA-seq (scRNA-seq) datasets. Using the BBKNN method (Polański *et al*, 2020), we calculated an integrated data manifold (as described in Materials and Methods) that represents gene expression profiles of 233,638 single-cells from 61 human individuals (ILD $n = 32$, controls $n = 29$) from all three studies (Fig 2A and B; Reyfman *et al*, 2019)—Chicago cohort: ILD $n = 9$, controls $n = 8$; Nashville cohort: ILD $n = 20$, controls $n = 10$; and Munich cohort: ILD $n = 3$, controls $n = 11$. Cell type identities were then annotated manually as described above (Fig 2C; Dataset EV2).

Next, we performed cell type-specific differential gene expression analysis accounting for demographic covariates across all 61 individuals. As a proof of concept, we first identified genes significantly associated with gender. The top hits were genes located on one of the sex chromosomes, demonstrating the validity of our approach (Appendix Fig S2A and B). Next, we identified genes differentially expressed between end-stage lung fibrosis and control tissue accounting for age, gender and study cohort (see Dataset EV4a for a full list of differential gene expression for health status stratified by cell type or by meta-cell type in Dataset EV4b). Gene expression changes in disease were most similar in cell types within the respective epithelial, mesenchymal, and leukocyte lineages (Fig 2D). This means that, for instance, the up- and downregulated genes in fibroblasts are more likely to be also regulated in other mesenchymal cells as in leukocytes and vice versa. Despite small differences in sequencing depth, differentially expressed genes showed very good agreement between the three independent patient cohorts (Fig 2E–H). The upregulated *KRT17* gene (Fig 2F) in alveolar epithelial cells was recently defined as a marker for the novel aberrant basaloid cells in IPF (Adams *et al*, 2020; Habermann *et al*, 2020). These fibrosis-specific cells feature a cellular senescence signature (Kobayashi *et al*, 2020), including high expression of *CDKN2A* (encoding for p16; Fig 2F). We also corroborate previous studies by showing that in fibroblasts (Fig 2G) the expression levels of *DIO2*, encoding for the thyroid hormone activating enzyme iodothyronine deiodinase (Yu *et al*, 2017), and the circulating CXCL14 (Jia *et al*, 2017; Rodriguez *et al*, 2018) chemokine are increased. In macrophages (Fig 2H), the normal alveolar macrophage phenotype that is marked by high FABP4 expression is replaced by an ILD-associated cell state that features high expression of *SPP1* (Osteopontin) (Morse *et al*, 2019), which we termed activated AM. These cells also express higher levels of the CCR2 ligand CCL7, which is a chemoattractant potentially involved in the recruitment of fibrocytes and profibrotic macrophages (Moore *et al*, 2005; Osterholzer *et al*, 2012, 2013). These examples manifest that integration of scRNA-seq

                                                    

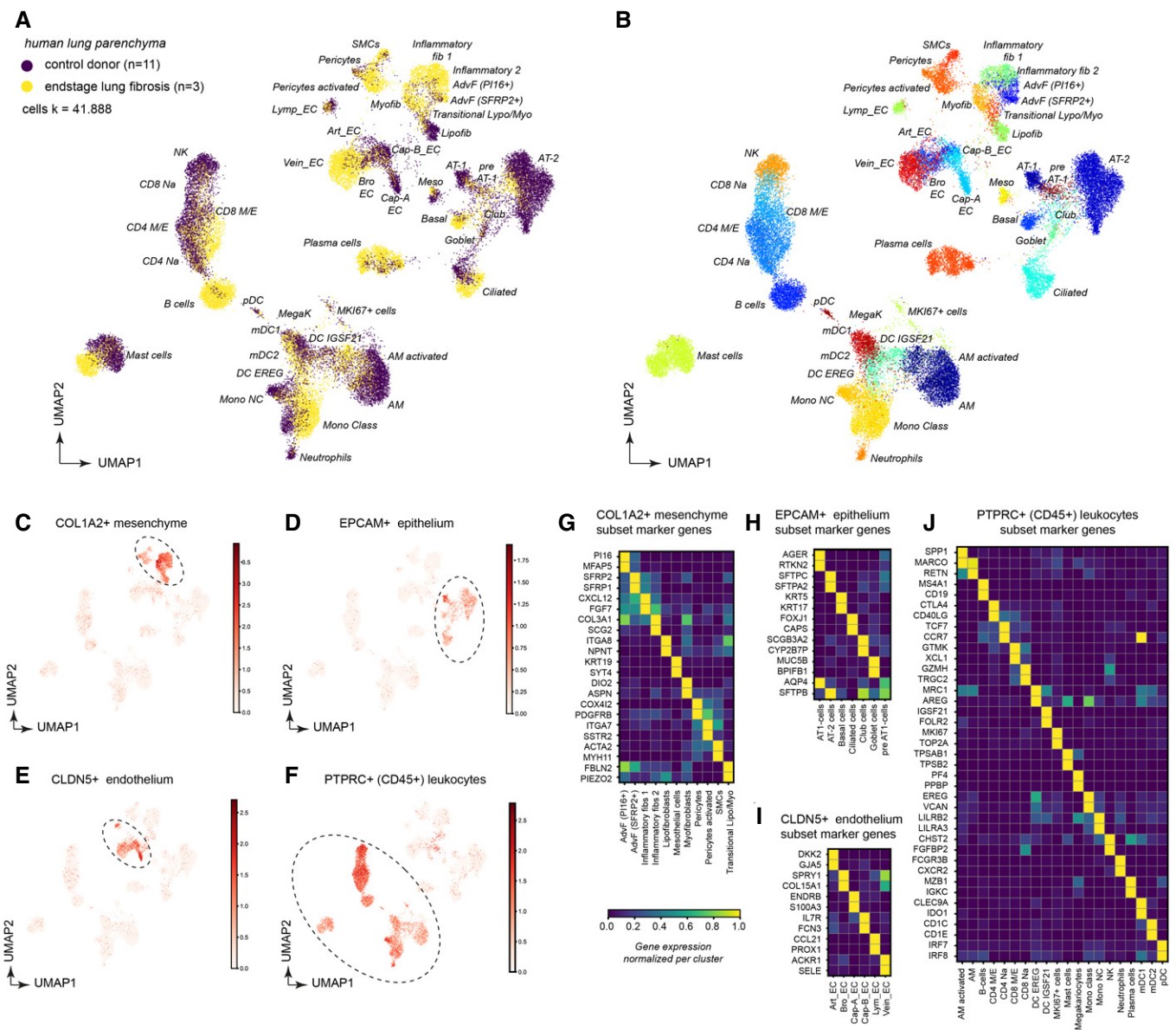

**Figure 1. Single-cell analysis of human lung parenchyma reveals 45 distinct cell type identities and their marker genes in ILD.**

A, B Dimension-reduced single-cell transcriptomic data are visualized through Uniform Manifold Approximation and Projection (UMAP). The color code illustrates the disease status (A) and cell type identity (B) (see Dataset EV3 for abbreviations).

C–F The indicated marker genes were used to select clusters for subsetting into stromal cells (C), epithelial cells (D), endothelial cells (E), and leukocytes (F).

G–J The heatmaps show the relative gene expression levels for the indicated marker genes for the indicated stromal (G), epithelial (H), endothelial (I), and leukocyte (J) cell types.

datasets facilitates highly powered and robust differential gene expression analysis, which represents a valuable resource to the research community.

Differences in cell type frequencies between individual samples can be caused by true biological differences, as well as differences in cell isolation protocols and scRNA-seq platforms used. Indeed, we observed large variance in cell population frequencies across cohorts and disease conditions (Appendix Fig S2C). After performing dimension reduction using PCA, we observed larger variation in principal components one and two among ILD patients compared to

control samples, indicating increased heterogeneity in disease (Fig 3A). Principal component two separated samples obtained from control donors and ILD patients across all three cohorts (Fig 3B). This observation motivated us to ask if cell type frequencies alone could distinguish ILD samples from controls. Therefore, we trained a random forest model based on the cell type proportions to predict disease status. This model achieved a mean accuracy of 83% derived from fivefold cross-validation (Fig 3C), demonstrating that the cell type frequencies showed robust differences in disease. Most important for the models prediction accuracy were changes in

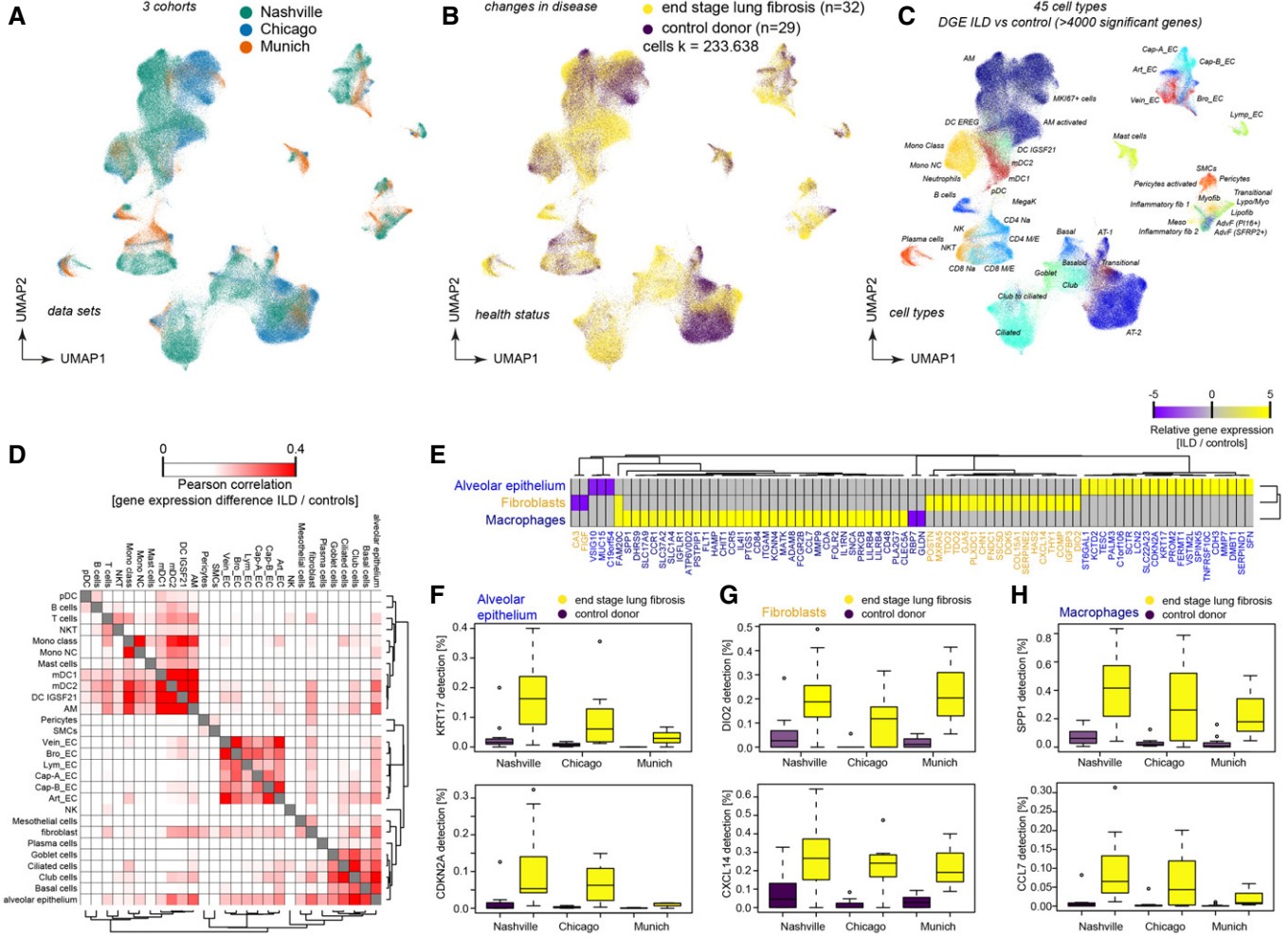

**Figure 2. Multi-cohort single-cell data reveal reproducible transcriptional changes for > 40 cell types.**

A–C  Dimension-reduced single-cell transcriptomic data are visualized through multiple Uniform Manifold Approximation and Projections (UMAPs). The colors illustrate (A) patient cohorts, (B) disease status, (C) and cell type identity (see Table S3 for abbreviations).

D  Differential gene expression between end-stage lung disease patients and controls across cohorts was compared for the indicated cell type identities. The color code demonstrates similarities of gene expression changes calculated by Pearson correlation of the t-value coefficient, which represents differences in the likelihood of detection for any gene between health and disease.

E  The heatmap shows the top 79 genes differentially expressed in the indicated cell type identities.

F–H  The box plots illustrate differences in mRNA detection for the indicated genes between tissues from end-stage lung fibrosis patients in (F) alveolar epithelial cells, (G) fibroblasts, and (H) macrophages when compared to control tissue. The boxes represent the interquartile range, the horizontal line in the box is the median, and the whiskers represent 1.5 times the interquartile range (Chicago cohort: ILD n = 9, controls n = 8; Nashville cohort: ILD n = 20, controls n = 10; Munich cohort: ILD n = 3, controls n = 11).

frequency of disease induced cell states as well as several parenchymal cell identities (Fig 3D). As expected, the decrease of AT1 and AT2 cells was important for the predictions. Interestingly, the top importance score in the model was achieved by the recently discovered aberrant basaloid cells (Adams et al, 2020; Habermann et al, 2020), suggesting that this cell state is indeed very disease-specific (Fig 3D).

To leverage the power of bulk RNA-seq data archived in public databases, we used our ILD single-cell atlas to determine possible cell type frequency changes in such datasets. A recent study used quantitative micro-CT imaging and tissue histology on biopsies to stratify lung tissue of idiopathic pulmonary fibrosis (IPF) patients

into different stages (IPF stage 1–3) marked by increasing extent of fibrotic remodeling (lower alveolar surface density and higher collagen content) (McDonough et al, 2019). Thus, the RNA-seq profiles of these staged patient samples presumably depict disease progression within patients. We calculated enrichment of our cell state signatures across the three stages of IPF progression and observed significant changes of many cell types already in early stage IPF-1, which still harbors more alveolar cell identities compared to the more advanced stages IPF-2 and IPF-3 (Fig 3E). This included the myofibroblast signature (Fig 3F) that was clearly upregulated early in progression as well as the aberrant basaloid cells. Other cell signatures, such as the plasma cells, showed a gradual increase

from IPF1 to IPF3, while for instance the increase in ciliated cell frequency was observed only from IPF stage 2 onwards (Fig 3F). Increases in airway cell frequencies with advanced stages are likely the consequence of the well described "bronchiolization" of the distal lung with metaplastic epithelial cells in IPF. Importantly, our analysis identifies cell state shifts and cell frequency changes that precede this bronchiolization. The appearance of the aberrant basaloid cells together with activated fibroblast states peaked already at the IPF-1 stage, indicating that these represent early events in disease progression. Notably, also several immune cell types, including B and T lymphocytes, were increased already in IPF-1.

In summary, we computationally integrated over 200 thousand single-cells from three independent lung fibrosis patient cohorts. Differential gene expression in lung fibrosis was robustly replicated and validated across cohorts and thus serves as a powerful resource to investigators studying ILD pathogenesis and progression, e.g., for the dissection of bulk RNA-seq profiles as demonstrated. We show that cell state changes as well as cell type frequency changes co-occur during disease progression and can be decoupled using single-cell analysis.

**Human lung bronchoalveolar lavage fluid proteomes reflect changes in disease activity**

Transcriptional changes are not always correlated with protein abundance, in particular if proteins are secreted (Angelidis *et al*, 2019). However, some of the cell state changes described by our single-cell analysis may be reflected in the proteomic composition of the bronchoalveolar lavage fluid (BALF), which is accessible for sampling during bronchoscopic examination of patients. Here, we used state-of-the-art mass spectrometry for in-depth analysis of BALF proteomes of a large ILD and non-ILD patient cohort (Fig 4A). Our ILD cohort included eight groups of patients (Fig 4B), with a diagnosis of idiopathic pulmonary fibrosis (IPF, $n = 16$), hypersensitivity pneumonitis (HP/EAA, $n = 8$), cryptogenic organizing pneumonia (COP, $n = 11$), idiopathic non-specific interstitial pneumonia (NSIP, $n = 10$), smoking-associated respiratory bronchiolitis ILD (RB-ILD, $n = 3$), Sarcoidosis ($n = 22$), unclassifiable ILDs (other ILDs, $n = 25$), or non-ILD conditions such as lung cancer or COPD (non-ILD, $n = 29$) (see Dataset EV5 for clinical features). Of note, the majority of lavage fluids from patients in this cohort were collected during evaluation of initial diagnosis of ILD and thus rather represent early disease. Nonetheless, some patients already had severely reduced lung function. From 124 patients (95 ILD and 29 non-ILD) that passed quality control criteria, we quantified a median number of 835 proteins per individual patient, resulting in a total of 1,513 unique proteins that were quantified in at least 20 patients (Fig 4B and Dataset EV6). This is a very good depth of analysis given that BALF is difficult to analyze by mass spectrometry because of the high dynamic range of protein copy numbers present due to plasma protein leakage. To illustrate a correlation of the extent of plasma protein leakage with the depth of proteome analysis in our BALF samples, we quantified the mass fraction of the top 100 abundant proteins in patient plasma in the BALF proteomes. The number of detected proteins in BALF inversely correlated with the relative proportion of plasma proteins present (Appendix Fig S3A); however, this was not strongly associated with plasma LDH, which

serves as a marker for tissue damage in patient blood. Using a quantitative comparison of proteins detected in both the ILD patient plasma and BALF proteomes revealed proteins that are detected in plasma proteomes but are quantitatively enriched in BALF, suggesting they are produced locally in the lung and then transpire to the plasma (Appendix Fig S3B).

To better define the proteins that are major constituents of the epithelial lining fluid (ELF) of the lung (rather than tissue leakage proteins), we also performed a quantitative comparison of BALF content and total tissue proteomes from 11 end-stage ILD tissue biopsies (Schiller *et al*, 2017). Proteins detected in both tissue and fluid proteomes were scored as either a "tissue leakage" protein or "epithelial lining fluid" protein based on their enrichments in the respective compartments (Appendix Fig S3C). Indeed, proteins specific for secretory epithelial cells such as Club and AT2 cells had a significantly higher ELF enrichment score as proteins specific to non-secretory AT1 cells. Similarly, we found that secreted proteins had a higher score than transmembrane proteins and cytoplasmic proteins (Appendix Fig S3D). We identified 7 proteins that were significantly associated with clinical parameters, had a high ELF score, and were also detected in patient plasma by mass spectrometry (Appendix Fig S3E). Mapping the expression of these genes on the individual cell type identities illustrates their specific expression patterns (Appendix Fig S3F).

A Fisher's exact test showed that 285 BALF proteins with high coefficient of variation (CV) across patients (Appendix Fig S4A) were significantly enriched for gene annotations such as "secreted", "plasma lipoprotein", "antimicrobial", "nucleosome", "intermediate filament", and "extracellular matrix" (Dataset EV7), indicating that these categories are regulated across patient groups. Principal component analysis revealed clusters of patients with heterogeneous diagnosis that were either significantly enriched in complement, coagulation proteins and plasma lipid transport proteins, or showed higher levels of antimicrobial proteins and histones, pointing toward the involvement of an inflammatory response driven by neutrophil extracellular traps (NETs) in these patients (Appendix Fig S4B and C). Correlation analysis of the protein expression profiles revealed that in fact many proteins were co-expressed across patients, revealing co-regulated protein modules that were enriched for distinct signatures, including macrophage specific proteins such as the Scavenger receptor cysteine-rich type 1 protein M130 (CD163) and the Complement C1q subcomponent subunit C (C1QC), wound healing factors, such as the ECM proteins Tenascin-C, Fibronectin, Collagen type 6 and Periostin, as well as lipid transport, complement and coagulation proteins such as Apolipoprotein B-100 and Complement component C7, or antimicrobial defense and neutrophil chemotaxis factors, including granulocyte specific proteins such as S100-A8, S100-A9, Cathelicidin antimicrobial peptide (CAMP) and Myeloperoxidase (MPO) (Appendix Fig S4D).

In correlation analysis, we identified associations of these protein signatures with 33 individual clinical measurements per patient, including various lung function parameters and plasma LDH (Dataset EV8 and Appendix Figs S5 and S6). We identified biomarker signatures by Pearson correlation of the clinical parameters with proteins that were quantified in BALF in at least 20 patients (Dataset EV6), revealing highly significant correlations of distinct sets of proteins with several lung function parameters,

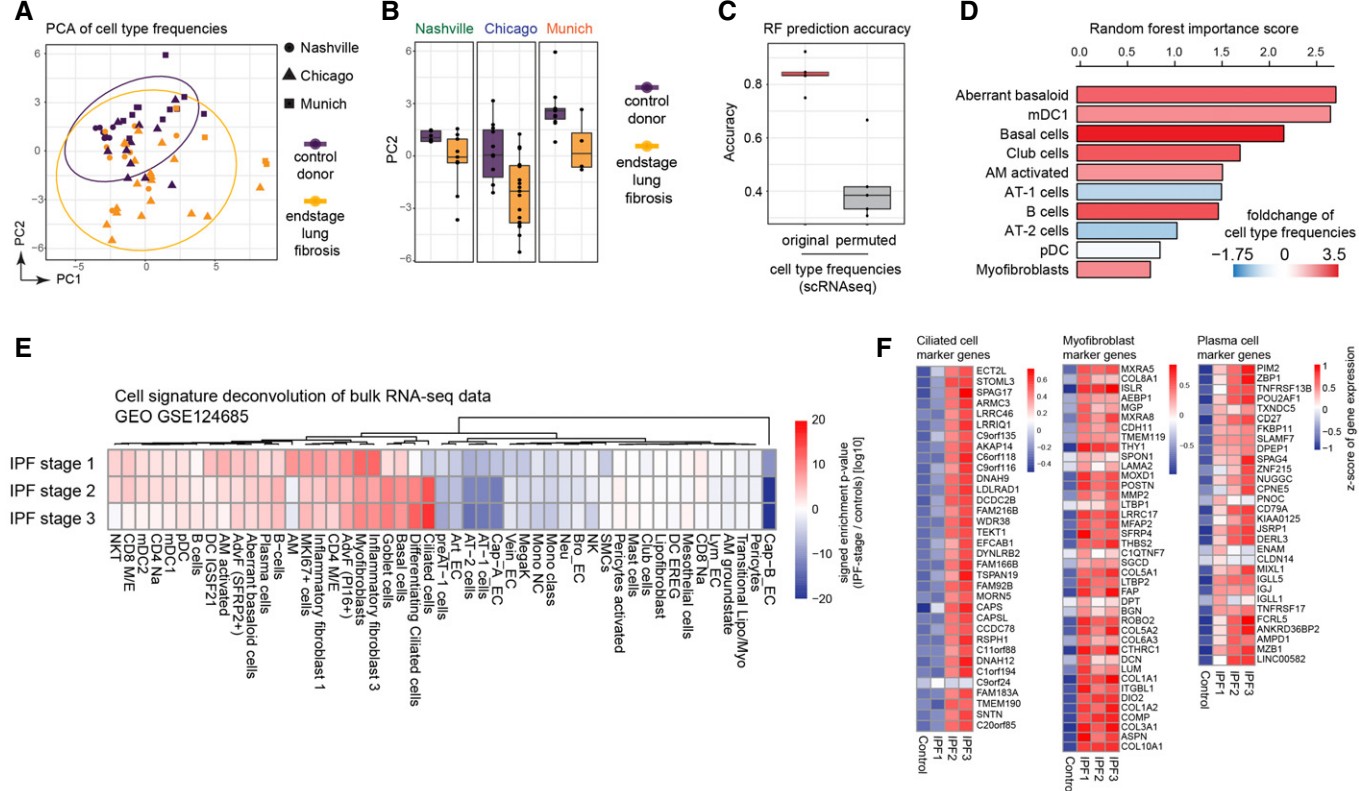

**Figure 3. Disease progression alters cell type frequencies.**

A   Principal components one (*x*-axis) and two (*y*-axis) illustrate the cell type frequencies in reduced dimensions. The shape of each point corresponds to the study cohort, and the points are colored by disease phenotype.

B   Box plots display principal component two across ILD patients (orange) and control samples (purple) for the three study cohorts. The boxes represent the interquartile range, the horizontal line in the box is the median, and the whiskers represent 1.5 times the interquartile range (Chicago cohort: ILD *n* = 9, controls *n* = 8; Nashville cohort: ILD *n* = 20, controls *n* = 10; Munich cohort: ILD *n* = 3, controls *n* = 11).

C   Box plot depicts random forest model prediction accuracies derived from fivefold cross-validation using the original and permuted labels. The boxes represent the interquartile range, the horizontal line in the box is the median, and the whiskers represent 1.5 times the interquartile range (*n* = 5).

D   Barplot shows the random forest importance scores for the top ten most informative features.

E   The heatmap shows changes of the indicated cell type signatures in published bulk RNA-seq data (GEO GSE124685) across different histopathological stages that represent increasing extent of fibrosis from stage 1–3, as determined by quantitative micro-CT imaging and tissue histology (McDonough *et al*, 2019). Samples used in this study were 10 IPF and 6 control patients.

F   The heatmaps show *z*-scores for the individual marker genes of the indicated cell types across IPF stages and controls.

including Diffusing Capacity For Carbon Monoxide (DLCO) or plasma levels for lactate dehydrogenase (pLDH) (Fig 4C). Most proteins that we found increased in BALF of patients with high pLDH were also associated with lower lung function (Appendix Fig S6A), and top outliers remained significant after accounting for patient age (Appendix Fig S6B–E). Because LDH is released during tissue damage and transpires to the blood, its levels in blood plasma are clinically used as a marker of ongoing cell death in tissues. We hypothesized that BALF proteins with correlation to pLDH in human patients represent a lung injury signature. A comparison of the human pLDH signature with BALF proteomes from mice after bleomycin injury revealed similar outlier proteins across species including the injury marker Tenascin-C (Appendix Fig S6F). Using 1D annotation enrichment analysis (see Materials and Methods), we confirmed that the pLDH correlation revealed protein changes in human patient BALF proteomes that were highly similar to the ones

that can be observed upon a defined acute lung injury in the bleomycin mouse model (Appendix Fig S6G).

To maximize the utility of this densely phenotyped cohort, we used principal component analysis to derive a single measure of "meta" lung function representing the combination of multiple lung function parameters (Fig 4D). Next, to account for potential confounding demographic variables, we performed linear multivariate regression analysis associating protein expression levels with the meta lung function variable accounting for age and gender (Fig 3E, Dataset EV9).

In summary, we analyzed BALF proteomes from 124 patients and correlated protein expression with an extensive set of clinical parameters, which represents, to the best of our knowledge, the most comprehensive characterization of this body fluid so far. Further, we identified co-regulated protein modules that were associated with patients lung function and current injury status,

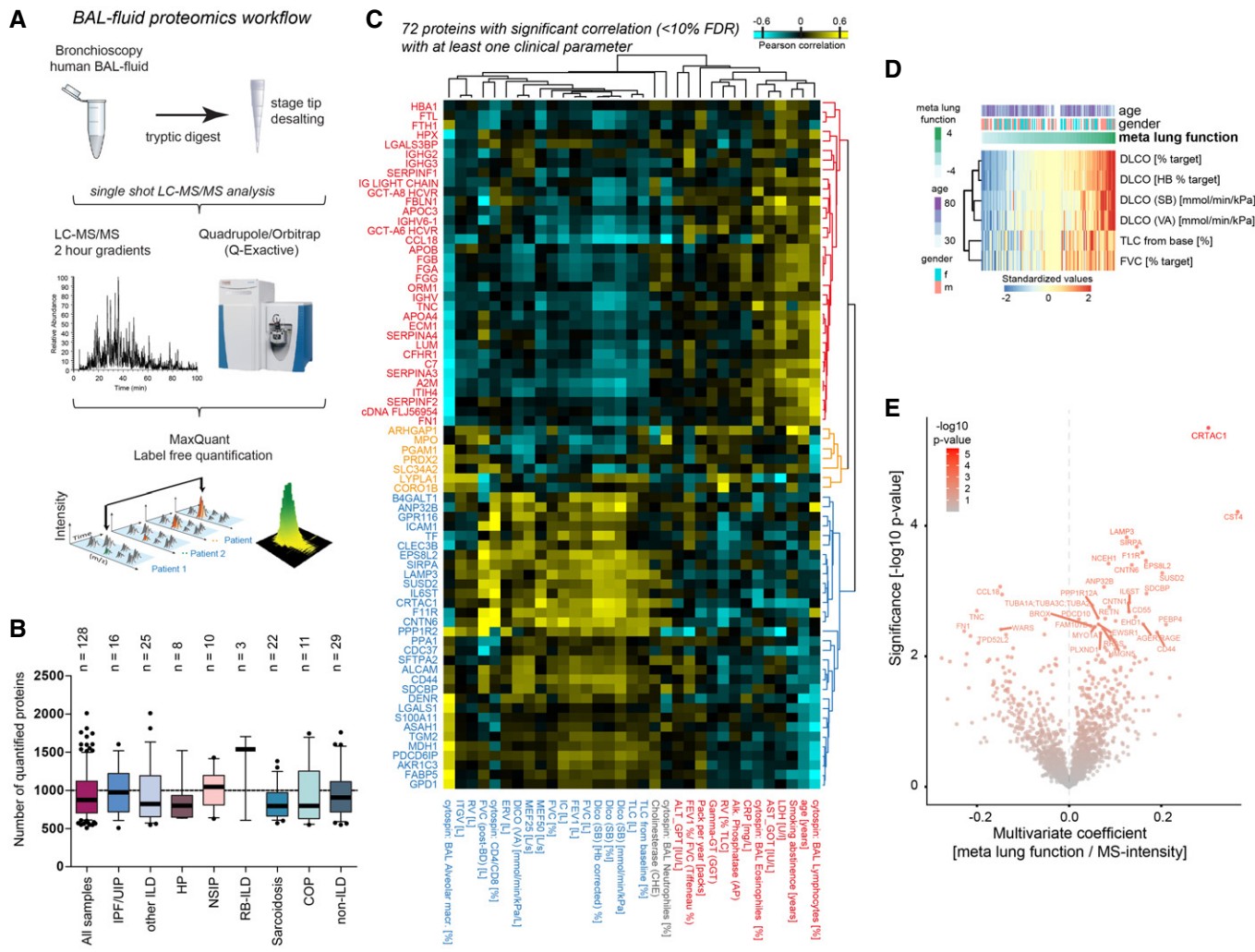

**Figure 4. Human lung bronchoalveolar lavage fluid proteome changes correlate with clinical parameters.**

A Proteomics workflow.

B The box plots show the number of proteins quantified (*y*-axis) across various diagnoses (*x*-axis). The mean and 10–90 percentiles are shown; the dotted line marks 1,000 proteins (IPF *n* = 16, HP/EAA *n* = 8, COP *n* = 11, NSIP *n* = 10, RB-IL *n* = 3, Sarcoidosis *n* = 22, other ILDs *n* = 25, non-IL, *n* = 29).

C The heatmap shows the correlation coefficients between proteins (rows) and clinical parameters (columns).

D The heatmap illustrates the computationally derived meta lung function variable combining multiple lung function parameters.

E The volcano plot shows the multivariate regression coefficients (*x*-axis) and the $-\log_{10}$ *P*-value (*y*-axis) for BALF protein abundance with the meta lung function.

suggesting that some of these protein signatures could be used to monitor acute or subclinical exacerbations of ILD patients.

**Correspondence of fluid proteins with cell state and frequency changes in the lung**

Next, we aimed to explain quantitative changes in BALF protein signatures with the cell state changes analyzed by single-cell RNA-seq. We first deconvoluted the diagnosis-specific protein biomarker signatures in the BALF proteomes and evaluated the relative contribution of cell types/states. Mean intensity *z*-scores of proteins across different diagnostic groups were tested for enrichment of cell type-specific transcriptional signatures (Fig 5A). Markers of several pro-fibrogenic cell types including fibroblast subsets, pericytes,

plasma cells, and mesothelial cells were strongly increased in protein measurements of COP, NSIP, HP/EAA, and IPF compared to non-ILD controls, confirming the power of BALF proteomics to correctly score fibrogenic remodeling in the patients. Interestingly, RB-ILD and Sarcoidosis samples were similar to non-ILD controls for this signature, which is consistent with their distinct histopathology that does not involve strong interstitial fibrosis. While RB-ILD protein analysis featured very strong enrichment for proteins specific to airway basal, ciliated, and goblet cells, the same airway protein signature was depleted in patients with COP, NSIP, and HP/EAA but not IPF (Fig 5A).

Similarly, we found strong associations between cell type/state signatures and the Pearson correlation of most clinical parameters with the protein measurements (Fig 5B). For instance, the

myofibroblast-specific proteins quantified in patient BALF tended to be negatively correlated with lung function (DLCO) (Fig 5C), and the number of alveolar macrophages in BAL cytospins tended to be negatively correlated with proteins (mostly antibodies) secreted by plasma cells into the BALF (Fig 5D).

Next, we aimed to understand the relationship between cell type-specific transcriptional changes and distinct signatures in the bronchoalveolar lavage fluid. To accomplish this aim, we integrated the results from two multivariate regression analyses: (i) protein associations with meta lung function and (ii) cell type-specific RNA

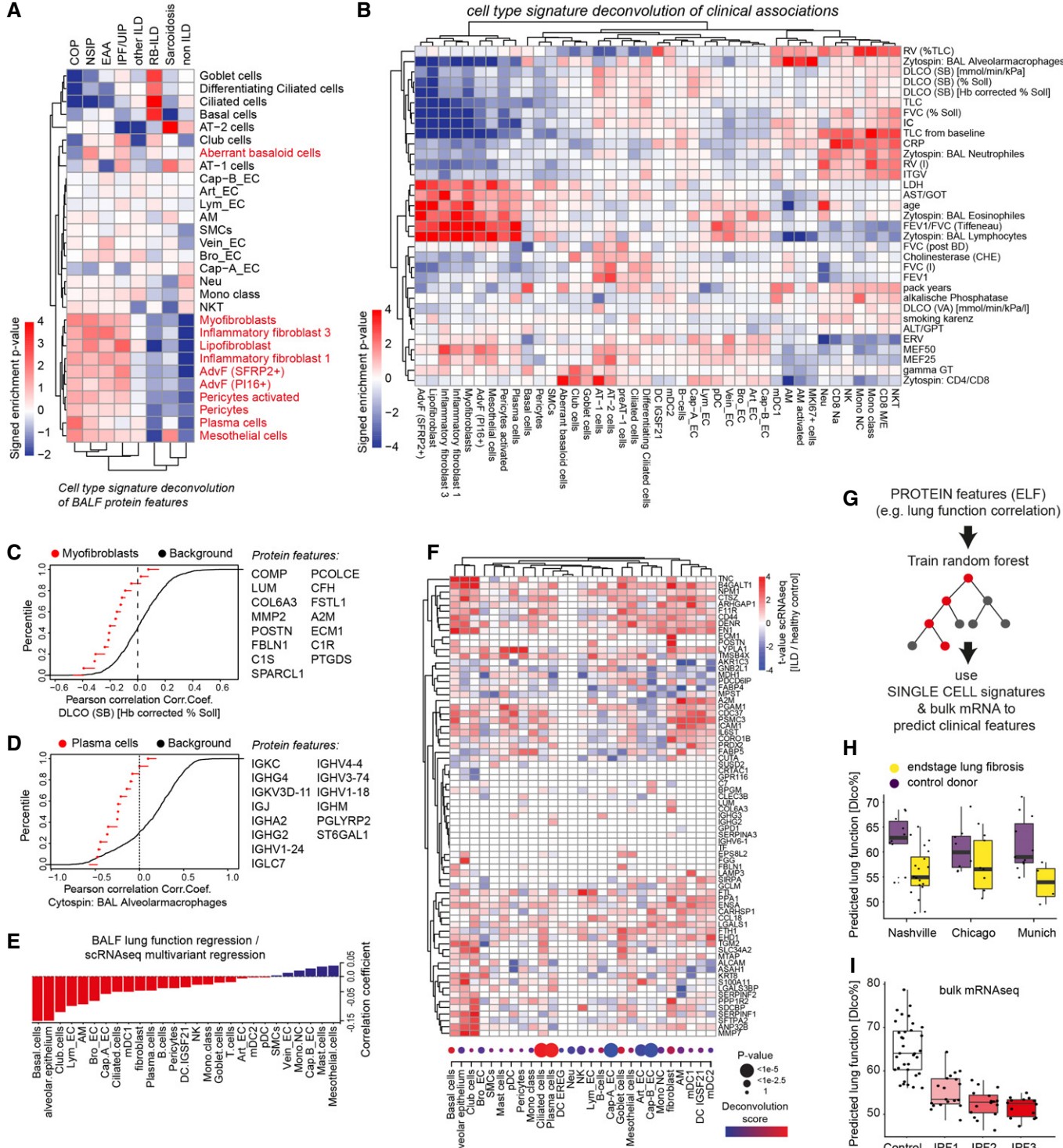

**Figure 5.**

**Figure 5. Protein signatures in BALF predict lung function decline and the corresponding cellular changes.**

A, B    The heatmaps show the deconvolution scores of cell types across BALF samples grouped by diagnosis (A) and indicated clinical parameters (B).

C, D    Empirical cumulative density plots depict the distribution of correlation coefficients for (C) Myofibroblast markers (red points) with DLCO and (D) Plasma cell markers (red points) with % alveolar macrophages in BAL in comparison to all background proteins (black line).

E       Barplot displays coefficients derived from correlating lung function associations at the protein level with cell type-specific ILD associations at the RNA level.

F       The heatmap illustrates gene expression changes associated with lung fibrosis across indicated cell types (columns) for selected BALF protein biomarkers (rows). The dotplot visualizes the frequency changes of the indicated cell types inferred from the deconvolution of bulk mRNA data of ILD samples compared to controls (GSE47460). Samples used in this study from the LTRC $n = 254$ ILD patients and $n = 108$ controls.

G       The protein features in BALF were used to train a random forest algorithm to predict lung function. The model was tested on transcriptional signatures from single-cell RNA-seq data to correctly predict reduced lung function in end-stage lung fibrosis when compared to controls.

H, I    Box plots show predicted lung function changes (DLCO%) in the three single-cell RNA-seq cohorts (Chicago cohort: ILD $n = 9$, controls $n = 8$; Nashville cohort: ILD $n = 20$, controls $n = 10$; Munich cohort: ILD $n = 3$, controls $n = 11$) in (H) and published bulk RNA-seq of IPF samples from different histopathological stages (GEO GSE124685) (control $n = 35$, IPF1 $n = 19$, IPF2 $n = 16$, IPF3 $n = 14$) in (I). The boxes represent the interquartile range, the horizontal line in the box is the median, and the whiskers represent 1.5 times the interquartile range.

associations with ILD status. Correlation of the *t*-values derived from both regressions revealed that the greatest correspondence between protein signatures associated with meta lung function and expression changes occured in club and basal cells as well as the alveolar epithelium (Fig 5E). However, the measured bulk protein profiles in BALF can be affected by two types of alterations: (i) changes in cell type frequency and (ii) changes in gene expression. Therefore, we inferred cell type frequency changes by performing deconvolution analysis on a large bulk expression dataset containing ILD patients and healthy controls from the Lung Tissue Research Consortium (LTRC; GSE47460). The top regulated BALF proteins were often altered both on gene expression and cell type frequency levels (Fig 5F), which importantly can be distinguished using scRNA-seq data.

Finally, to test if we could successfully transfer information from the proteomics modality into the scRNA-seq data modality, we applied machine learning. A random forest was trained on the protein quantification data to predict lung function (DLCO) using a set of protein features which (i) showed high correlation with lung function (DLCO) and (ii) had the corresponding transcript detected in the scRNA-seq data. Next, the trained model was applied to *in silico* bulk scRNA-seq data with mRNA expression mapped to proteins (Fig 5G), which then correctly predicted the direction of lung function changes in the three single-cell RNA-seq cohorts (Fig 5H). In addition, we applied an analogous approach to published bulk RNA-seq data of IPF samples from different histopathological stages determined by quantitative micro-CT imaging and tissue histology (GEO GSE124685) (McDonough *et al*, 2019; Fig 5I). Our model successfully predicted continuous lung function decline along the histopathological IPF disease stages, indicating that the BALF protein biomarker profiles discovered in this study quantitatively reflect cell state changes during disease progression.

Stratification along the protein and cell type-specific *t*-values revealed the expected inverse correlations between protein association with meta lung function and upregulation of the corresponding gene in ILD for several cell types, including KRT5+ basal cells and alveolar epithelial cells (Fig 6A–F). To histologically validate some of the most significantly regulated proteins in BALF and put these into the context of the cell types identified by scRNA-seq, we performed immunofluorescence analysis (Fig 6G–J). Increased expression of the extracellular matrix protein Tenascin-C, which is a known marker in tissue repair and was also upregulated in mouse lung injury (Appendix Fig S6F), was found in IPF tissue sections in

KRT5+ basal cells in the "bronchiolized" distal lung (Fig 6G). As predicted by the scRNA-seq analysis, these cells also co-expressed increased levels of the 14-3-3 protein sigma (SFN), which is an adaptor protein with possible functions in epithelial cell growth and regulation of the p53 pathway (Yang *et al*, 2008; Fig 6H and I). SFN has recently also been described as a marker in a novel transitional stem cell state that transiently appears during lung regeneration (Strunz *et al*, 2020; Kobayashi *et al*, 2020). This transitional stem cell state is highly similar to the aberrant basaloid state discovered in IPF (Adams *et al*, 2020; Habermann *et al*, 2020), which also expresses increased amounts of SFN. The transitional stem cell state in lung injury repair is also characterized by increased levels of KRT8 expression (Strunz *et al*, 2020), and interestingly, we found that cells with increased levels of KRT8 in metaplastic epithelial areas in IPF lung also co-expressed increased levels of SFN compared to controls (Fig 6J). Thus, in summary, our in situ validation suggests that the observed negative correlation of TNC and SFN protein in BALF is the consequence of a general upregulation of these markers in several epithelial states within the metaplastic and "bronchiolized" epithelium in IPF.

In summary, our cross-modality analysis serves as proof of concept that cell state and frequency changes in diseased organs can transpire into predictive body fluid protein signatures that can be analyzed by mass spectrometry with high precision.

## A disease associated pericyte state in ILD

One of the protein biomarkers in BALF that showed a significant negative correlation with lung function was the Complement factor H-related protein 1 (CFHR1) (Fig 7A). CFHR1 activates the complement system *in vivo* via competitive antagonism with Complement factor H (CFH), which is a soluble inhibitor of complement (de Jorge *et al*, 2013). Increased activation of complement has been reported in IPF and was discussed to perpetuate epithelial injury (Meliconi *et al*, 1990; Gu *et al*, 2014; Pankita & Pandya, 2014). Thus, to follow this important lead we used our single-cell data to shed light on the cellular source of ILD-specific upregulation of CFHR1.

Expression of CFHR1 was exclusively detectable in COL1A2+ stromal cells (Appendix Fig S7). In the COL1A2+ cells, the CFHR1 expression was further limited to an activated PDGFRB+ pericyte state that was mainly found in ILD patients (Fig 7B). Differential gene expression analysis within the pericyte population indeed identified CFHR1 as a top regulated gene (Fig 7C), together with many

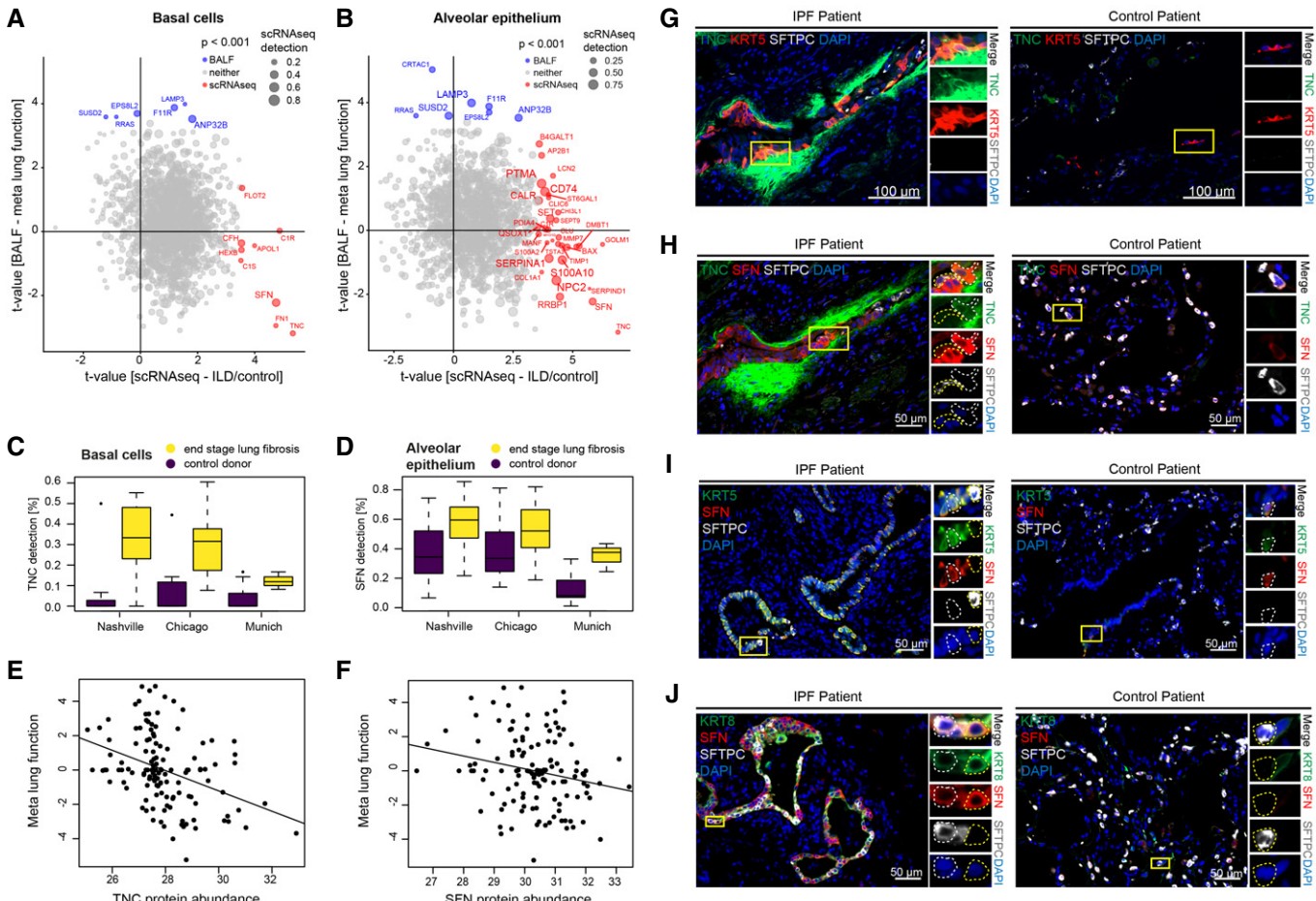

**Figure 6. Cell type-specific transcriptional ILD signatures translate into protein signatures associated with lung function in the BALF.**

A, B   Scatter plots stratify genes based on the protein lung function associations (y-axis) and cell type-specific ILD associations (x-axis). The size of the dots represents the detection level of each gene in the corresponding cell type. Colors highlight genes with marginal associations at the protein and RNA levels.

C, D   The box plots illustrate differences in mRNA detection for the indicated genes between tissues from end-stage lung fibrosis patients in (C) basal cells and (D) alveolar epithelial cells when compared to controls. The boxes represent the interquartile range, the horizontal line in the box is the median, and the whiskers represent 1.5 times the interquartile range (Chicago cohort: ILD n = 9, controls n = 8; Nashville cohort: ILD n = 20, controls n = 10; Munich cohort: ILD n = 3, controls n = 11).

E, F   The scatter plots show the positive correlation of the indicated proteins in BALF (MS-intensity, x-axis) with meta lung function (y-axis).

G–J   Immunofluorescence analysis of the indicated proteins and cell type markers in IPF (n = 3) and control samples (n = 3).

other genes that were significantly enriched for several gene categories, including "chemokine activity", "G-protein-coupled receptor", and "complement regulatory" genes (Fig 7D).

The observed upregulation of the G-protein-coupled receptor gene *SSTR2* has previously been associated with ILD (Schniering *et al*, 2019a), but its cell type-specific expression patterns were unclear. To identify gene programs in pericytes that are associated with high *SSTR2* expression, we performed gene–gene correlation analysis across single-cells. To mitigate the impact of sparse counts on correlation measures, we aggregated cells into small clusters of transcriptionally similar cells before calculating correlation following previous work (Iacono *et al*, 2019). The correlation analysis revealed a large cluster of genes that was strongly associated with SSTR2 in pericytes, including *CFHR1, CFH, CXCL2/3, CD36,* and *YAP1* (Fig 7E). The disease-specific expression of SSTR2 in

PDGFRB+ pericytes was confirmed using immunofluorescence microscopy (Fig 7F) and immunohistochemistry (Fig 7G). SSTR2+/PDGFRB+/DES− pericytes were found around remodeled vessels that had a thickened layer of DES+/PDGFRB+ smooth muscle cells in ILD. PDGFRB+/DES− pericytes were negative for SSTR2 in control lungs with normal thickness of the smooth muscle cell layer (Fig 7F). We quantified the SSTR2 immunohistochemistry signal in 79 tissue sections from 53 ILD patients and 26 control patients, and correlated this signal with the severity of fibrotic remodeling using an Ashcroft scoring (Fig 7H). The SSTR2 levels were strongly associated with high Ashcroft scores, indicating that the SSTR2+/CFHR1+ pericyte state is correlated with the severity of fibrosis.

Gene–gene correlation analysis revealed that the transcriptional regulator Yap1 was strongly associated with *SSTR2* expression (Fig 7I). However, upstream regulator analysis did not reveal a clear

Yap1 target gene signature within the set of SSTR2 correlated genes (Fig 7J). This prediction instead pointed toward a STAT1/NFKB-driven inflammatory signature that could be consistent with the many immune-associated genes (e.g., CXCL2/3) co-expressed with *SSTR2* and *CFHR1* (Fig 7J).

To predict the hierarchy of gene expression during the state change of pericytes in ILD, we performed a pseudotemporal modeling analysis of this differentiation trajectory (Fig 7K–M). Our model confirmed the gradual upregulation of *SSTR2* and *CFHR1* with their correlating genes (Fig 7E) concurrent with downregulation of several pericyte marker genes such as *PDGFRB* (Fig 7M). Thus, in

summary we have discovered a novel ILD-associated pericyte state that may affect pathogenesis via its influence on local complement activation and immune cell recruitment. The appearance of this novel *SSTR2*+ pericyte state is furthermore reflected by increased levels of *CFHR1* protein in the lavage fluid.

**CRTAC1 is a novel peripheral protein biomarker of AT2 cell health status in the lung**

The BALF protein with the highest and most significant positive association in our multivariate regression meta lung function

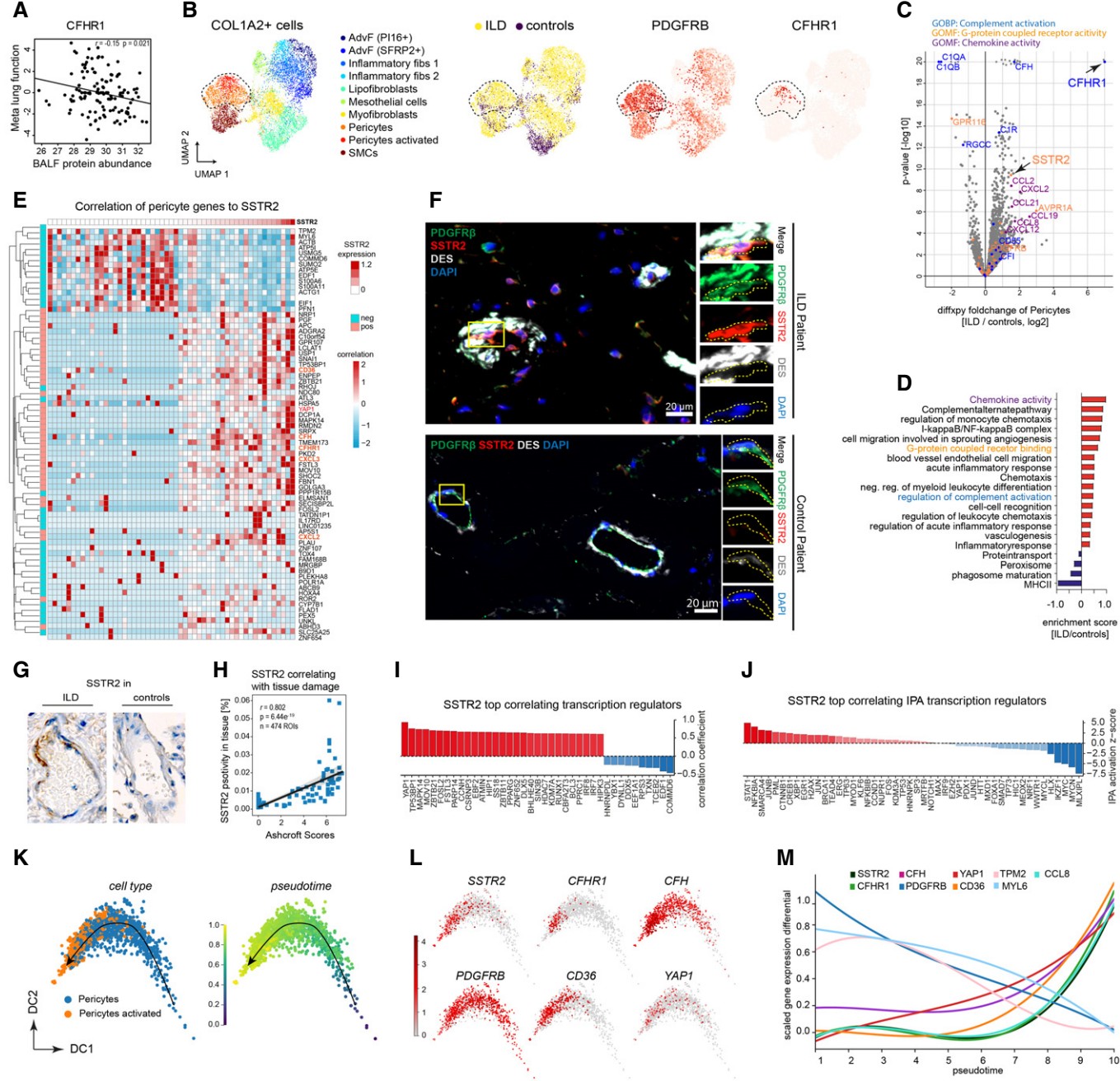

**Figure 7.**

◄

**Figure 7.  An SSTR2+/CFHR1+ pericyte state correlates with progression of fibrotic remodeling.**

A    The scatter plots show the positive correlation of CFHR1 in BALF (MS-intensity, *x*-axis) with meta lung function (*y*-axis).

B    The UMAPs show the subset of COL1A2+ mesenchymal cells in the integrated scRNA-seq dataset, with the pericytes highlighted by the dotted line. From left to right, the colors represent the cell type identities, disease status, expression of the pericyte markers *PDGFRB* and *CFHR1*, respectively.

C    The volcano plot depicts the fold changes (*x*-axis) and the $-\log_{10}$ *P*-values (*y*-axis) derived from the differential gene expression analysis using diffxpy between ILD and controls within the subset of PDGFRB+ pericytes (outlier values truncated).

D    The bar graphs show the top regulated gene categories after enrichment analysis using GO terms and UniProt keywords.

E    The heatmap shows the expression of genes most strongly associated with SSTR2 expression within the pericytes (left color bar indicates significant correlation or anti-correlation).

F    Immunofluorescence analysis of SSTR2, the pericyte cell type marker PDGFRB and a marker for smooth muscle cells DESMIN in IPF (*n* = 3) and control samples (*n* = 3).

G    Representative images of immunohistochemistry analysis of SSTR2 protein expression in tissue regions (*n* = 474) from ILD patients (*n* = 53) and control patients (*n* = 26).

H    Correlation of immunohistochemistry signal of SSTR2 with Ashcroft scores.

I    The bar graph shows the genes most strongly correlated with SSTR2 belonging to the GO category "transcription regulators". The dotted line marks a correlation coefficient of zero.

J    The bar graph shows the top correlated transcriptional regulators, predicted by ingenuity pathway analysis (IPA) for the SSTR2 gene-gene correlations. The dotted line marks a correlation coefficient of zero.

K    Diffusion map of pericytes colored by cell type and inferred pseudotime represents pericyte differentiation.

L    Diffusion map of pericytes colored by the gene expression of indicated genes.

M    The line plot illustrates smoothed expression levels of the indicated genes across the pericyte pseudotime differentiation trajectory.

analysis was the cartilage acidic protein 1 (*CRTAC1*), whose function in the lung is currently unknown (Figs 4E and 8A). Our single-cell atlas revealed specific expression of *CRTAC1* in lung lymphatic endothelium, airway club cells, and most prominently in alveolar type-2 epithelial (AT2) cells (Fig 8B, Appendix Fig S7B). On the whole body level, the mRNA expression of *CRTAC1* was highest in the lung (Fig 8C). Expression of *CRTAC1* in alveolar epithelial cells was consistently downregulated in ILD samples compared to controls in all three patient cohorts analyzed by single-cell RNA-seq (Fig 8D). Also re-analysis of published bulk transcriptomes confirmed a highly significant downregulation of *CRTAC1* mRNA in the lung of ILD patients compared to healthy controls and COPD patients (Fig 8E).

To identify gene programs within AT2 cells that are associated with *CRTAC1* expression, we performed gene–gene correlation analysis within SFTPC+ AT2 cells. Positively and negatively correlated genes were identified and those correlations were reproducible across all three cohorts (Fig 8F). Genes positively correlated with CRTAC1 across all cohorts were significantly enriched for categories that are consistent with a normal AT2 cell identity, including surfactant genes, and cholesterol biosynthesis genes. The anti-correlated genes were enriched for immune and inflammatory processes, as well as extracellular matrix and intermediate filament genes (Fig 8G).

Analysis of transcriptional regulators revealed that the expression of the transcriptional activator C/EBP-delta (CEBPD) was highly correlated with *CRTAC1* expression (Fig 8H). *CEBPD* expression and activity is induced by glucocorticoids and has a role in AT2 cell differentiation during lung development (Breed *et al*, 1997; Berg *et al*, 2002). Interestingly, the levels of CRTAC1 in isolated human AT2 cells increase upon differentiation with glucocorticoids (Ballard *et al*, 2010), which are known to be essential for alveolar maturation in lung development (Gerber, 2015). Thus, we speculate that the downregulation of CRTAC1 in AT2 cells of ILD patients may hint at currently uncharacterized changes in glucocorticoid signaling in these cells. We also performed upstream regulator analysis in ingenuity pathway analysis (IPA) to predict the activity of transcriptional regulators based on the correlated or anti-correlated gene

profiles (Fig 8I). This analysis identified ETV5 as a potential regulator of the CRTAC1 correlated gene program in AT2. This is in line with the fact that ETV5 has been shown to be essential for AT2 cell maintenance *in vivo*, as deletion of *Etv5* from mouse AT2 cells produced gene and protein signatures characteristic of differentiated alveolar type I (AT1) cells (Zhang *et al*, 2017). Thus, our finding here is consistent with the notion that CRTAC1 is associated with a normal healthy and highly differentiated AT2 cell state.

The most strongly anti-correlated transcriptional regulator to the *CRTAC1*-associated gene programs in AT2 cells was the transcription factor SOX4 (Fig 8H). SOX4 is regulated by various pathways, including TGF-beta signaling, and we recently described Sox4 together with Nupr1 as a candidate transcriptional regulator of AT2 cell differentiation upon lung injury in the mouse (Strunz *et al*, 2020). Interestingly, the IPA upstream regulator analysis predicted high activity of NUPR1 also in AT2 cells with low expression of *CRTAC1* (Fig 8I). NUPR1 plays a role in cell stress responses, including DNA damage repair and regulation of the cellular senescence program. The co-regulation of *SOX4* and *NUPR1* expression and activity in both mouse and human AT2 cells in health and disease suggests that this program may have important functions in AT2 cell de-differentiation. We show that *CRTAC1* expression is strongly anti-correlated to this de-differentiation program.

We modeled the de-differentiation of human AT2 cells in ILD by deriving a pseudotime trajectory using all SFTPC+ AT2 cells and the aberrant basaloid cells across all three cohorts (Fig 8J). The pseudotime trajectory showed a gradual increase of aberrant basaloid cell markers starting already in still SFTPC+ AT2 cells and peaking in then SFTPC−/SOX4+ aberrant basaloid cells, which also expressed genes such as SFN and TNC that we had found to be increased in the BALF, and the cytoskeletal protein Cornifin-alpha (SPRR1A) (Fig 8K). Importantly, the downregulation of *CRTAC1* occurs in an early stage of this differentiation process. Even though *CRTAC1* expression in AT2 is specific to humans and not observed in mice, we find that the differentiation trajectory we modeled in human IPF is highly similar to a differentiation trajectory observed in mice after bleomycin injury (Fig 8L). The pseudotime trajectory of mouse AT2

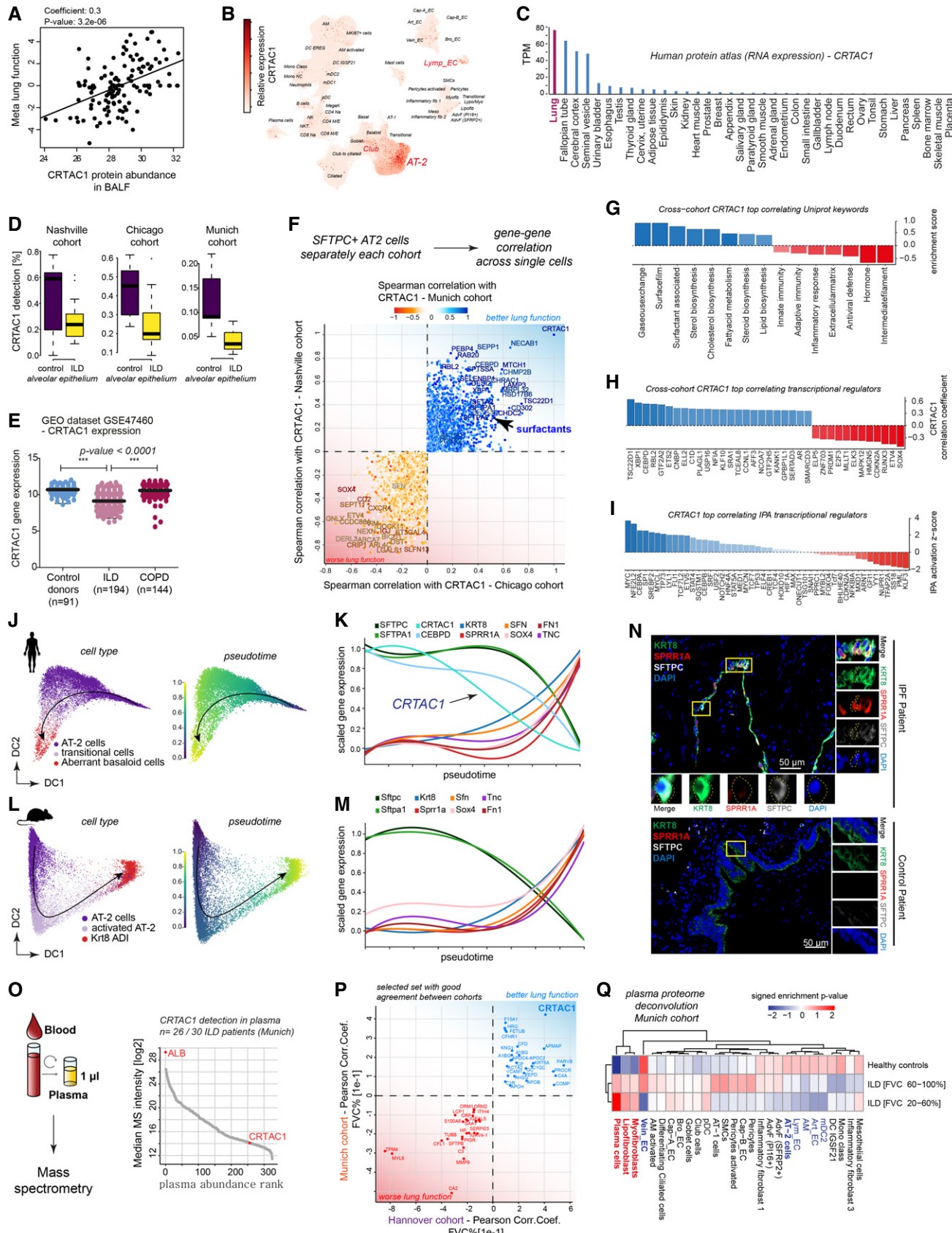

**Figure 8.**

**Figure 8. CRTAC1 protein abundance in BALF and plasma proteomes reports AT2 cell health.**

A   The scatter plots show the positive correlation of CFHR1 in BALF (MS-intensity, *x*-axis) with meta lung function (*y*-axis).
B   UMAP visualizes embedding of single-cells colored by gene expression for *CRTAC1*, which is specifically expressed in alveolar type-2 (AT2), Club and lymphatic endothelial (Lymp_EC) cells.
C   Relative expression level of *CRTAC1* across human organs.
D   The box plots illustrate differences in mRNA detection for *CRTAC1* in alveolar epithelial cells from fibrosis patients compared to control samples across the three indicated patient cohorts (Chicago cohort: ILD *n* = 9, controls *n* = 8; Nashville cohort: ILD *n* = 20, controls *n* = 10; Munich cohort: ILD *n* = 3, controls *n* = 11). The boxes represent the interquartile range, the horizontal line in the box is the median, and the whiskers represent 1.5 times the interquartile range.
E   Relative gene expression levels of *CRTAC1* in GSE47460. Dots represent average expression in the tissue of individual patients. The line represents the mean, and error bars show SD. *CRTAC1* is significantly downregulated in ILD but not COPD patients (*P*-value < 0.0001) (one-way ANOVA) (control donors *n* = 91, ILD *n* = 194, COPD *n* = 144).
F   For each single-cell cohort, the gene–gene correlations with *CRTAC1* within the SFTPC+ AT-2 cells were calculated. The indicated genes were selected based on their common direction of correlation across cohorts.
G   The bar graph shows the gene categories most strongly correlated with *CRTAC1* based on "UniProt keywords". The dotted line marks a correlation coefficient of zero.
H   The bar graph shows the gene categories most strongly correlated with *CRTAC1* belonging to the GO category of "transcription regulators". The dotted line marks a correlation coefficient of zero.
I   The bar graph shows the top correlated transcriptional regulators, predicted by ingenuity pathway analysis (IPA) for the *CRTAC1* gene–gene correlations. The dotted line marks a correlation coefficient of zero.
J   Diffusion map of human AT2 cells colored by cell type identity and inferred pseudotime.
K   The line plot illustrates smoothed expression levels of the indicated genes across the human AT2 pseudotime trajectory.
L   Diffusion map of mouse AT2 cells colored by cell type identity and inferred pseudotime.
M   The line plot illustrates smoothed expression levels of the indicated genes across the (Niu *et al*, 2019) mouse AT2 pseudotime trajectory.
N   Immunofluorescence analysis of SPRR1A, KRT8 as well as SFTPC in IPF (*n* = 3) and control samples (*n* = 2).
O   A high-throughput experimental workflow for plasma proteomics (Niu *et al*, 2019) allowed for profiling of two independent cohorts of ILD patients (Munich, *n* = 30 and Hannover, *n* = 81; healthy age-matched controls, *n* = 30). All proteins quantified in plasma are shown, ranked by their abundance measured by mass spectrometry (MS-intensity).
P   The indicated proteins from the plasma analysis were selected based on their common direction of correlation with patient lung function in two independent patient cohorts with distinct clinical characteristics.
Q   The heatmap shows the predicted relative contribution of lung cell types to the association of protein biomarker signatures in plasma with lung function (forced vital capacity—FVC). Patients were split in two groups, one with a mild decline in lung function [FVC 60–100%] and one with severe loss of lung function [FVC 20–60%] and compared to healthy age-matched controls.

cells toward the Krt8+ alveolar differentiation intermediate (ADI) cell state (Strunz *et al*, 2020) shows a similar upregulation of *Sox4, Sfn, Krt8, Fn1, Tnc*, and *Sprr1a* as observed in the human trajectory (Fig 8M). This strongly suggests that aberrant basaloid cells in IPF may be generated from AT2 in an attempt for regenerative repair. Indeed, we find KRT8+/SPRR1A+/SFTPC− cells in close proximity to SFTPC+ cells in areas of limited fibrotic remodeling, which may represent early stage disease (Fig 8N).

Clinical decisions are often based on blood biomarker analysis (Geyer *et al*, 2017). To extend our analysis to the plasma proteome, we made use of a recently established high-throughput plasma proteomics workflow (Geyer *et al*, 2017, 2019; Niu *et al*, 2019) and generated plasma proteomes from two independent cohorts of ILD patients (Munich, *n* = 30 and Hannover, *n* = 81; healthy age-matched controls, *n* = 30; see Dataset EV10 for clinical characteristics and Dataset EV11 for plasma protein quantification). We were able to robustly detect CRTAC1 by mass spectrometry in > 80% of the plasma samples (Fig 8O). The Hannover cohort included more patients with better lung function on average, with samples taken mainly at time of initial diagnosis, while the Munich cohort contained patients closer to end-stage disease. Thus, by construction, we did not expect a perfect match of these two cohorts. Upon correlation with forced vital capacity (FVC %), we identified a shared panel of proteins in both cohorts that were either positively or negatively associated with the lung function outcome (Fig 8P). As expected, CRTAC1 showed a positive correlation with lung function in both patient cohorts.

Finally, we investigated the relative contribution of cell types/ states to the protein biomarker signatures in plasma that correlated with lung function (Fig 8Q). We divided the patients into two groups representing mild and severe disease based on lung function (FVC %) and compared these two groups with healthy controls. We observed a gradual increase of proteins potentially derived from lung fibroblast subsets and plasma cells as well as a gradual reduction of proteins potentially derived from lung endothelial cells, alveolar macrophages, AT2 cells, and mDC2 (Fig 8Q).

In conclusion, our data and analysis suggest that plasma proteomes harbor protein biomarker signatures that report the status of cell states in health and disease. Here, we demonstrated that the AT2-derived CRTAC1 protein in ILD patient plasma and BAL fluid correlates with lung function and reports the loss of AT2 cell identity during disease progression.

# Discussion

The field of single-cell genomics has rapidly evolved and with the increasing availability of cell atlases is now moving toward the mechanistic characterization of pathogenesis and disease progression, as well as translational applications in medicine. We can conceptualize interindividual variance within patient cohorts with a model in which patients at different stages of a disease progression trajectory will have the diseased organs in different characteristic states. Body fluids potentially contain a composite representation of these disease stage-specific differences of cell type/state proportions within affected organs in the form of proteins and possibly cell-free DNA. We must deconvolute these composite signatures in order to make predictions about cell and tissue level changes in the patient. In this work, we

explore this idea and predict the cellular sources of protein biomarker profiles in body fluids. We envision training machine learning algorithms with large datasets of matched single-cell genomic and fluid proteomic or sequencing readouts derived from longitudinal measurements in patient cohorts. This will enable the development of new automated tools for clinical decision making (Walsh *et al*, 2019) and drug monitoring (Maher *et al*, 2019) for future predictive and interceptive medicine (Rajewsky *et al*, 2020).

In idiopathic pulmonary fibrosis (IPF), the most common form of lung fibrosis, the progressive replacement of lung parenchyma with scar tissue leads to respiratory failure with a median survival time of 2–4 years after diagnosis. Current models of disease pathogenesis propose that a combination of repetitive (micro)injuries to susceptible alveolar epithelial cells (AEC) with an aberrant repair response causes pathological interactions of AEC with fibroblasts and subsequent accumulation of scar tissue (Kropski & Blackwell, 2019). Human genetic data and pre-clinical models show that epithelial injury can drive subsequent fibrosis, with a combination of genetic predisposition and aging thought to be related to the failed regenerative response in IPF. The recent discovery of a transitional stem cell state that transiently appears in normal lung regeneration but persistently accumulates in lung fibrosis enables a new perspective on pathogenesis of IPF (Verheyden & Sun, 2020; Strunz *et al*, 2020; Kobayashi *et al*, 2020). This transitional stem cell state in mice is highly similar to the aberrant basaloid cells in IPF (Adams *et al*, 2020; Habermann *et al*, 2020) and features the expression of several pro-fibrogenic factors that may activate mesenchymal cells. We show that the main source of CRTAC1 in the human body is the alveolar AT2 cell and that CRTAC1 is downregulated early in a de-differentiation trajectory of AT2 cells toward the aberrant basaloid state. Our data suggests that this cellular transition is reflected in plasma proteomes by declining abundance of CRTAC1. We therefore propose that CRTAC1 protein levels in plasma and lavage fluids specifically report the AT2 cell health status. This novel biomarker is thus a promising candidate for future prospective trials in various settings, including monitoring the degree of distal lung involvement during virus induced pneumonia, as currently seen in the COVID-19 pandemic. Supporting our hypothesis, a recent preprint illustrates that CRTAC1 is downregulated in plasma of hospitalized COVID-19 patients compared to SARS-CoV-2 negative controls (Filbin *et al*, 2020).

Perivascular cells have been shown to be key contributors to organ fibrosis, including the lung (Kramann *et al*, 2015), and a pericyte to myofibroblast transition in lung fibrosis has been proposed (Hung *et al*, 2019). As our single-cell analysis demonstrates, the previous functional studies of PDGFRB+ cells isolated from lungs underappreciated the heterogeneity of PDGFRB+ cells, which not only contain perivascular pericytes but also fibroblast populations and smooth muscle cells. In this work, we discover a highly disease-specific SSTR2+/CFHR1+ pericyte state that features a pro-inflammatory phenotype with expression of various chemokines. Interestingly, the expression of the potent complement regulatory factor CFHR1 in this cell state may explain the previously observed deregulation of complement in IPF. Further functional investigations on this novel pericyte state in lung fibrosis will shed light on its potential direct relevance in disease progression. Interestingly, visualizing radiolabeled somatostatin analogues targeting the *SSTR2* receptor have recently been proposed for the visualization of fibrotic changes in ILD (Lebtahi *et al*, 2006; Ambrosini *et al*, 2010; Schniering *et al*, 2019a).

Lung fibrosis patients experience highly diverse clinical courses, with progression often accelerated due to acute exacerbations (AE), associated with a high mortality (Collard *et al*, 2016). Efforts have been made to find predictive biomarkers for AE and disease outcome (Collard *et al*, 2010; Neighbors *et al*, 2018). For instance, elevated serum levels of AT2-derived SP-A and SP-D are associated with an increased risk of mortality in IPF (Greene *et al*, 2002; Kinder *et al*, 2009; Collard *et al*, 2010). CCL18 has been reported to be elevated in the serum and BALF in patients with lung fibrosis (Prasse *et al*, 2007), which we also confirm in this study. In fact, a recent pooled *post hoc* analysis of the CAPACITY and ASCEND studies identified CCL-18 as the most robust blood marker for disease progression in IPF (Neighbors *et al*, 2018). Nevertheless, such biomarkers are currently not clinically established and often it is unclear which cellular changes they represent. While early detection of AE is currently of major interest, there are also patients who present with clinical worsening without meeting the criteria for AE. A daily home spirometry study resulted in highly diverse lung function trajectories in IPF (Russell *et al*, 2016), suggesting that lung function diversity could also reflect different stages after epithelial lung injury with phases of decreased lung function (potentially being a phase of subclinical injury/exacerbation) followed by phases of slightly increased lung function (potentially being a phase of successful tissue repair).

Our machine learning analysis demonstrates that correspondence of fluid proteomes and single-cell transcriptomes can be used to correctly predict the direction of lung function changes across modalities. The BALF proteome signature used to train the random forest algorithm correctly predicted declining lung function across data modalities in a micro-CT staged histopathological progression from very early structural changes to complete fibrotic remodeling. This indicates that the protein features derived from this large heterogeneous patient cohort would likely also correctly report disease progression in a longitudinal setting. This justifies further development of this concept, which we hope will contribute to future clinical decision making. Our work has several limitations that prohibited us from fully completing this task. Since the cross-modal analysis was done on non-matched patient cohorts, it is currently difficult to assess the specificity of fluid proteome signatures for tissue level cell state changes, and to go beyond associative signatures. Thus, carefully designed longitudinal multi-modal analysis of animal models and patient cohorts will be required to train machine learning algorithms, in particular causal inference models, for future applications in predictive personalized medicine.

# Materials and Methods

### Human samples

Human samples of the Munich cohorts (tissue, BAL fluid and plasma) were obtained from the bioArchive of the Comprehensive Pneumology Center Munich (CPC-M). Written informed consent was obtained from all patients, and the study was approved by the local ethics committee of the Ludwig-Maximilians University of Munich, Germany (EK 333-10 and 382-10). ILD lung tissue for single-cell analysis was freshly obtained after lung transplantation at the University Hospital Munich and compared to lung tissue of

non CLD patients as tumor free, uninvolved lung tissue freshly obtained during tumor resections performed at the lung specialist clinic "Asklepios Fachkliniken Munich-Gauting". BAL fluid samples of the BAL fluid cohort and matched plasma samples were collected at the lung specialist clinic and included mainly first ILD evaluations. Plasma samples from an independent ILD cohort were obtained from patients seen in the ILD outpatient clinic of the Hospital of the Ludwigs Maximilian University in Munich during routine visits or as an inpatient during evaluation for lung transplantation. Plasma samples of the Hannover cohort were obtained as a cooperation within the German Center for Lung Research (DZL); patients gave written informed consent (DZL broad-consent), and the study was approved by the local ethics committee of the Medizinische Hochschule Hannover (2923-2015). For the histology sections of the SSTR2 histochemistry and Ashcroft scoring, the local ethics committee of the University Hospital Zürich approved the study (BASEC-No. 2017-01298), and informed consent was obtained from all patients. Informed consent was obtained from all subjects, and the experiments conformed to the principles set out in the WMA Declaration of Helsinki and the Department of Health and Human Services Belmont Report.

The diagnosis of IPF was made in accordance with the current guidelines (Raghu *et al*, 2015). All ILD diagnoses were made according to international guidelines and established criteria. Non-ILD patients of the BAL fluid cohort included patients who underwent BAL due to evaluation of asthma, COPD, lung cancer, hemoptysis, or chronic cough.

For transport from the operation room to the laboratory, lung tissue samples for single-cell analysis were stored in ice-cold DMEM-F12 media in thermo stable boxes. Tissue was processed for single-cell analysis with a maximum delay of 2 h after surgery.

## Lung tissue processing

Lung tissue was processed as previously described (Vieira Braga *et al*, 2019). Briefly, around 1.5 g of tissue per sample was manually homogenized into smaller pieces (~ 0.5 mm$^2$ per piece). Before tissue digestion, lung homogenates were cleared by washing excessive blood through a 40-μm strainer with ice-cold PBS. The tissue was transferred into enzyme mix consisting of dispase, collagenase, elastase, and DNase for mild enzymatic digestion for 1 h at 37°C while shaking. Enzyme activity was inhibited by adding PBS supplemented with 10% FCS. Dissociated cells in suspension were passed through a 70-μm strainer and pelleted. The cell pellet was resuspended in red blood cell lysis buffer and incubated shortly at room temperature to lyse remaining red blood cells. After incubation, PBS supplemented with 10% FCS was added to the suspension and the cells were pelleted. The cells were taken up in PBS supplemented with 10% FCS, counted using a Neubauer chamber, and critically assessed for single-cell separation and viability. Two-hundred and fifty thousand cells were aliquoted in PBS supplemented with 0.04% of bovine serum albumin and loaded for Drop-seq at a final concentration of 100 cells/μl.

## Single-cell sequencing using Drop-seq

Drop-seq experiments were performed largely as described previously (Macosko *et al*, 2015) with few adaptations during the single-

cell library preparation (Angelidis *et al*, 2019). Briefly, using a microfluidic polydimethylsiloxane device (Nanoshift), single-cells from the lung cell suspension were co-encapsulated in droplets with barcoded beads (ChemGenes). Droplet emulsions were collected for 15 min each before droplet breakage was performed using perfluorooctanol (Sigma-Aldrich). After breakage, beads were collected and the hybridized mRNA transcripts reverse transcribed (Maxima RT, Thermo Fisher). Unused primers were removed by the addition of exonuclease I (New England Biolabs). Beads were washed, counted, and aliquoted for pre-amplification with 12 PCR cycles (primers, chemistry, and cycle conditions identical to those previously described). PCR products were pooled and purified twice by 0.6× clean-up beads (CleanNA). Before tagmentation, cDNA samples were loaded on a DNA High Sensitivity Chip on the 2100 Bioanalyzer (Agilent) to ensure transcript integrity, purity, and amount. For each sample, 1 ng of pre-amplified cDNA from an estimated 1,000 cells was tagmented by Nextera XT (Illumina) with a custom P5 primer (Integrated DNA Technologies). Single-cell libraries were sequenced in a 100 bp paired-end run on the Illumina HiSeq 4000 using 0.2 nM denatured sample and 5% PhiX spike-in. For priming of read 1, 0.5 μM Read1CustSeqB (primer sequence: GCCTGTC CGCGGAAGCAGTGGTATCAACGCAGAGTAC) was used.

## Processing of single-cell data from Munich

For the single-cell data of human patients form the Munich cohort, the Drop-seq computational pipeline was used (version 2.0) as previously described (Macosko *et al*, 2015). Briefly, STAR (version 2.5.2a) was used for mapping (Dobin *et al*, 2013). Reads were aligned to the hg19 reference genome (GSE63269). For barcode filtering, we excluded barcodes with less than 200 detected genes. For further filtering, we kept the top barcodes based on UMI count per cell, guided by the number of estimated cells per sample. As we observed a certain degree of ambient RNA bias, we applied SoupX (Young & Behjati, 2020) to lessen this effect. The pCut parameter was set to 0.3 within each sample before merging the count matrices together. The merged expression table was then pre-processed further. A high proportion (> 10%) of transcript counts derived from mitochondria-encoded genes may indicate low cell quality, and we removed these unqualified cells from downstream analysis. Cells with a high number of UMI counts may represent doublets; thus, only cells with less than 4000 UMIs were used in downstream analysis. Genes were only considered if they were expressed in at least three cells in the dataset (Dobin *et al*, 2013).

## Analysis of single-cell data from Munich

The downstream analysis of the Munich single-cell data was performed using the Scanpy Package (Wolf *et al*, 2018), a python package for the exploration of single-cell RNA-seq data. Following the common procedure, the expression matrices were normalized using *scran*'s (Lun *et al*, 2016) normalization based on size factors which are calculated and used to scale the counts in each cell. Next log transformation was used via scanpy's pp.log1p(). Highly variable genes were selected as follows. First, the function pp.highly_variable_genes() was executed for each sample separately, returning the top 4,000 variable genes per sample. Next, we only considered a gene as variable if it was labeled as such in at least two samples, resulting

in a total of 15,096 genes which were further used for the principal component analysis. In an additional step to mitigate the effects of unwanted sources of cell-to-cell variation, we regressed out the number of UMI counts, percentage of mitochondrial DNA and the calculated cell cycle score using the function pp.regress_out().

For visualizing the whole Munich dataset, the UMAP was generated using 50 components as input for scanpy's tl.umap() with number of neighbors set to 10 and min_dist parameter to 0.4. To better align the data of the different patients and to account for possible batch effects, we used the python package bbknn() (batch balanced k nearest neighbors) (Polański et al, 2020) with the same number of components and neighbors. Louvain clustering was calculated with resolution 6. The whole lung parenchymal dataset was split into subsets for COL1A2+ mesenchymal cells, EPCAM+ epithelial cells, CLDN5+ endothelial cells, and PTPRC+ leukocytes. New UMAP embeddings of these subsets were calculated until clear separation of cluster identities was achieved that allowed for identification of cell states by exploring the highest expressed markers per cluster explored via tl.rank_genes_groups() and manual assessment of known marker gene expression.

## Computational data integration of single-cell data

To improve statistical power, to ensure generalization across cohorts and to achieve a more balanced ratio of diseased and healthy patients, our Munich single-cell RNA-seq dataset was combined with the filtered count matrices from the Chicago cohort (Reyfman et al, 2019) and the Nashville cohort.

Before combining these, the count matrices from Chicago and Nashville were processed separately. The normalization using scran and the log transformation of the two external datasets was performed as described for the Munich cohort. The effect of cell cycle, the percentage of mitochondrial reads, and the number of UMI counts were regressed out cohort-wise as well.

For a first lighter batch correction, we defined the list of variable genes in a way to decrease cohort-specific effect as follows. For both the Nashville and the Chicago data, we considered a gene as highly variable if it is labeled highly variable in at least three patients of the respective dataset. Next, the pre-processed count matrices from the three datasets were merged and genes retained their highly variable status if they were highly variable in at least two of the three cohorts, resulting in 3,854 variable genes.

The concatenated object was scaled with scanpy's pp.scale() function, and the principal components were calculated using the defined variable genes. As a second batch correction, we calculated the neighborhood graph using the bbknn package, defining the individual patients as batch key, five number of neighbors within batch and 40 components. As described for the Munich cohort, the whole combined object was subsetted and new embeddings were calculated in order to identify cell states.

## Differential gene expression analysis

To identify genes associated with ILD status in a cell type-specific manner, we applied the following procedure. The R statistical software was used for the analysis. Since the outcome of interest (ILD status) varies at the sample ($n = 61$) as opposed to the cell level ($n = 233,638$), we framed the analysis as a likelihood of detection

problem across all samples. For each sample and cell type combination, we calculated the likelihood of detection for each gene as the average number of cells with more than one count. As the likelihood of detection represents a probability and is bounded between 0 and 1, values were square-root transformed. Next, we used multivariate regression to model the probabilities of detection. The square-root transformed detection probability was used as the dependent variable and the ILD status or gender as the explanatory variable accounting for the total number of UMI counts, total number of cells, study indicator, age and gender as covariates. The resulting $t$ and $P$-values for the coefficient describing the ILD status or gender were used in downstream analysis. Due to the relatively low number of pericytes, differential expression analysis within pericytes was performed using diffxpy (https://github.com/theislab/diffxpy).

## Cell type signature enrichment analysis

To infer cell type frequency changes from bulk transcriptomics or proteomics data, we applied signature enrichment analysis. We defined cell type signatures as sets of genes with significant cell type-specific expression as defined in Dataset EV2. Next, we statistically evaluated enrichment of each signature in a ranked list of fold changes or correlation coefficients using the Kolmogorov–Smirnov test. The signed $P$-value score represents the $-\log_{10} P$-value of the Kolmogorov–Smirnov test signed by the effect size. Negative and positive values represent depletion and enrichment of the given signature in the ranked list, respectively.

## Random forest prediction

To integrate scRNA-seq with BALF data, we used a random forest as implemented in the R *randomForest* package. First, BALF expression data were quantile normalized and scaled. Next, only features with an absolute correlation coefficient greater than 0.2 with lung function and present in the scRNA-seq data were used to train a random forest to predict lung function. Then, *in silico* bulk scRNA-seq was calculated by taking the mean expression count of each gene across all cells for all samples. Finally, the *in silico* bulk data were quantile normalized and scaled before feeding it into the trained model to predict lung function. In addition, we applied the analogous approach with the bulk RNA-seq data of IPF samples from different histopathological stages determined by quantitative micro-CT imaging and tissue histology (GEO GSE124685).

## Gene–gene correlation analysis

The gene–gene correlation analysis followed the approach described by Iacono et al (Iacono et al, 2019). More precisely, the subsets of pericytes and AT2 cells were used for each analysis independently. Principal components were recalculated and subjected to Louvain clustering using a relatively high-resolution parameter in order to derive a relatively large number of transcriptionally similar cell clusters. Next, expression across these cell clusters was averaged. Pearson correlation coefficients were calculated across these meta-cell cluster averages to derive the gene-gene correlations. Averaging the expression across small numbers of cells mitigates the impact of sparse counts at the single-cell level and leads to increased correlation values.

## Pseudotime analysis

To model the gene expression dynamics underlying the ILD-specific pericyte and AT2 trajectories, the following analysis was performed. Data were subset to pericytes or AT2 cells, and each subset was analyzed independently. Principal components were recalculated. Subsequently, a diffusion map was derived based on the principal components and pseudotime coordinates were calculated using the Scanpy function *sc.tl.dpt()*. To visualize gene expression along the inferred trajectory, pseudotime coordinates were divided into equally sized bins and expression values were averaged within each bin. A smooth fit was calculated across these expression averages and displayed using line plots.

## Clinical parameters

For all patients included in the final analysis, clinical information was collected at the time of BAL fluid procedure or when plasma was taken, respectively. Clinical parameters included demographics (age, gender, smoking status, pack years, smoking abstinence, lung function [forced vital capacity (FVC) (% pred.), FVC (l), FVC (post-broncholysis), expiratory reserve volume (ERV), forced expiratory volume in 1 s (FEV1) (l), FEV1/FVC (%), inspiratory capacity (IC) (l), total lung capacity (TLC) (l), TLC from baseline, residual volume (RV), RV (%TLC), diffusing capacity of the lung for carbon monoxide (DLCO) (VA) (mmol/min/kPa/l), DLCO (SB) (% pred.), DLCO (SB) (Hb corrected, % pred.), DLCO (SB) (mmol/min/kPa), mean expiratory flow (MEF) 25, MEF50, intrathoracic gas volume (ITGV)], laboratory values [cholinesterase, alkaline phosphatase, C-reactive protein, alanine-aminotransferase (ALT), aspartate-amino-transferase (AST), gamma-glutamyltransferase (GGT), LDH].

## BAL procedure

BALF was collected from 141 patients undergoing bronchoscopy from January 2013 until March 2016 at the Lungenfachklinik Gauting in Munich, Germany. Most of the patients underwent bronchoscopy due to ILD evaluation. BAL was performed with standard technique. In brief, 100–200 ml of sterile saline (0.9% NaCl) was instilled into the right middle lobe or the lingula in 20-ml injections which were each immediately aspirated. Cells of the BAL were analyzed by cytospin analysis. The remaining cell-free BAL fluid was immediately stored at −80°C and transferred to the BioArchive of the CPC-M. For mass spectrometry, only the cell-free BAL fluids were analyzed. Of the 141 patients, only 124 passed quality control and were included in the analysis (95 ILD and 29 non-ILD).

## Mass spectrometry

The BAL fluid depleted from cells was subjected to mass spectrometry analysis. Proteins were precipitated from 300 µl BAL fluid using 80% ice-cold acetone, followed by reduction and alkylation of proteins and overnight digestion into peptides using Trypsin and LysC proteases (1:100) as previously described (Schiller *et al*, 2015). Peptides were purified using StageTips containing a Poly-styrene-divinylbenzene copolymer modified with sulfonic acid groups (SDB-RPS) material (3 M, St. Paul, MN 55144-1000, USA) as previously described (Kulak *et al*, 2014). Approximately 2 µg of peptides were separated in 4 h gradients on a 50-cm long (75-µm inner diameter) column packed in-house with ReproSil-Pur C18-AQ 1.9 µm resin (Dr. Maisch GmbH). Reverse-phase chromatography was performed with an EASY-nLC 1000 ultra-high pressure system (Thermo Fisher Scientific), which was coupled to a Q Exactive Mass Spectrometer (Thermo Scientific). Peptides were loaded with buffer A (0.1% ($v/v$) formic acid) and eluted with a non-linear 240-min gradient of 5–60% buffer B (0.1% ($v/v$) formic acid, 80% ($v/v$) acetonitrile) at a flow rate of 250 nl/min. After each gradient, the column was washed with 95% buffer B and re-equilibrated with buffer A. Column temperature was kept at 50°C by an in-house designed oven with a Peltier element (Thakur *et al*, 2011), and operational parameters were monitored in real time by the SprayQC software (Scheltema & Mann 2012). MS data were acquired with a shotgun proteomics method, where in each cycle a full scan, providing an overview of the full complement of isotope patterns visible at that particular time point, is follow by up-to-ten data-dependent MS/MS scans on the most abundant not yet sequenced isotopes (top10 method) (Michalski *et al*, 2011). Target value for the full scan MS spectra was $3 \times 10^6$ charges in the $300-1,650$ $m/z$ range with a maximum injection time of 20 ms and a resolution of 70,000 at $m/z$ 400. The resulting mass spectra were processed using the MaxQuant software (Cox & Mann, 2008), which enabled label free protein quantification (Tyanova *et al*, 2016).

Plasma samples were prepared with the Plasma Proteome Profiling Pipeline (Geyer *et al*, 2016) automated on an Agilent Bravo liquid handling platform. Briefly, plasma samples were diluted 1:10 in ddH$_2$O and 10 µl were mixed with 10 µl PreOmics lysis buffer (P.O. 00001, PreOmics GmbH) for reduction of disulfide bridges, cysteine alkylation, and protein denaturation at 95°C for 10 min. Trypsin and LysC were added at a ratio of 1:100 micrograms of enzyme to micrograms of protein after a 5 min cooling step at room temperature. Digestion was performed at 37°C for 1 h. An amount of 20 µg of peptides was loaded on two 14-gauge StageTip plugs, followed by consecutive purification steps according to the PreOmics iST protocol (www.preomics.com). The StageTips were centrifuged using an in-house 3D-printed StageTip centrifugal device at 1,500 $g$. The collected material was completely dried using a SpeedVac centrifuge at 60°C (Eppendorf, Concentrator plus). Peptides were suspended in buffer A* (2% acetonitrile ($v/v$), 0.1% formic acid ($v/v$)) and shaking for 10 min at room temperature. Plasma peptides were measured using LC-MS instrumentation consisting of an Evosep One (Evosep), which was online coupled to a Q Exactive HF Orbitrap (Thermo Fisher Scientific). Peptides were separated on 15 cm capillary columns (ID: 150 µm; in-house packed into the pulled tip with ReproSil-Pur C18-AQ 1.9 µm resin (Dr. Maisch GmbH)). For each LC-MS/MS analysis, about 0.5 µg peptides were loaded and separated using the Evosep 60 samples method. Column temperature was kept at 60°C by an in-house-developed oven containing a Peltier element, and parameters were monitored in real time by the SprayQC software. MS data were acquired with data independent acquisition using a full scan at a resolution of 120,000 at $m/z$ 200, followed by 22 MS/MS scans at a resolution of 30,000.

## Mass spectrometry bioinformatic and statistical analyses

MS raw files of the plasma samples were analyzed by Spectronaut software [version 12.0.20491.10.21239 (Bruderer *et al*, 2015; Geyer

*et al*, 2016)] from Biognosys with default settings applied and were searched against the human UniProt FASTA database. Mass spectrometry raw files of the BALF samples were processed using the MaxQuant software (Cox & Mann, 2008) (*version 1.5.3.34*). As previously described (Schiller *et al*, 2015), peak lists were searched against the human UniProt FASTA database (November 2016), and a common contaminants database (247 entries) by the Andromeda search engine (Cox *et al*, 2011). Pearson correlation analysis, *t*-test statistics, ANOVA tests, or Fisher's exact test were performed using the GraphPad Prism 5 software. To identify proteins associated with lung function, the following procedure was used. Since a small fraction of samples contained missing values, principal component analysis, as implemented in the imputePCA() function of the missMDA R package, was used to derive a single "meta" lung function variable. Missing values in the BALF protein expression were imputed using the missForest() function of the missForest R package. Imputed values were used to model meta lung function accounting for covariates age and gender. The multivariate regression was implemented using the lm() function. *P*-values were adjusted for multiple testing using the R p.adjust() function. All other statistical and bioinformatics operations (such as normalization, data integration, annotation enrichment analysis, correlation analysis, hierarchical clustering, principal component analysis, and multiple-hypothesis testing corrections) were run with the Perseus software package (version 1.5.3.0 and 1.6.1.1.) (Tyanova *et al*, 2016).

## Immunofluorescence and microscopy

Formalin-fixed paraffin-embedded (FFPE) lung sections (3.5 μm thick) from ILD patients and controls were stained as previously described. In brief, after deparaffinization, rehydration, and heat-mediated antigen retrieval with citrate buffer (10 mM, pH = 6.0), sections were blocked with 5% bovine serum albumin for 1 h at room temperature followed by overnight incubation with the following primary antibodies at 4°C: rabbit anti-TNC (abcam, ab108930, 1:100), rabbit anti-SPRR1A (abcam, ab125374, 1:2,000), rabbit anti-SFTPC (Sigma-Aldrich, HPA010928 1:150), rabbit anti-SSTR2 (abcam, ab134152, 1:50), rabbit anti-YAP (abcam, ab205270, 1:500), rat anti-KRT8 (University of Iowa Hybridoma Bank, 1:200), mouse anti-PDGFRβ (Origene, TA506230, 1:50), mouse anti-SFTPC (Santa Cruz Biotechnologies, sc-518029, 1:50), chicken anti-Krt5 (BioLegend, Poly9059, 1:1,000), goat anti-SFN (abcam, ab77187, 1:250), goat anti-DES (Santa Cruz Biotechnologies, sc-7559, 1:100), and goat anti-CD45 (LifeSpan Biosciences, LS-B14248, 1:300). For visualization of stainings, the following secondary antibodies were used: donkey anti-rabbit Alexa Fluor 488 (Invitrogen, A-21206, 1:250), donkey anti-rat Alexa Fluor 488 (Invitrogen, A21208, 1:250), donkey anti-mouse Alexa Fluor 488 (Invitrogen, A21202, 1:250), goat anti-chicken Alexa Fluor 568 (Invitrogen, A11041, 1:250), donkey anti-rabbit Alexa Fluor 568 (Invitrogen, A10042, 1:250), donkey anti-rabbit Alexa Fluor 647 (Invitrogen, A31573, 1:250), goat anti-mouse Alexa Fluor 647 (Invitrogen, A21236, 1:250), and donkey anti-goat Alexa Fluor 647 (Invitrogen, A21447, 1:250). Tissue sections were additionally stained with 4',6-diamidino-2-phenylindole (DAPI, 10 mins at room temperature) to visualize cell nuclei, and tissue autofluorescence was blocked using the Vector TrueVIEW Autofluorescence Quenching Kit (Vector Laboratories).

### The paper explained

**Problem**
Future personalized interceptive medicine wants to assess early onset of disease in a pre-clinical stage when the patient is still feeling healthy. To work toward this goal, we must understand the changes to cellular circuits in patient organs early in disease. Furthermore, we must know how these changes are reflected in easy-to-access body fluids that can be used for straightforward longitudinal sampling and surveillance.

**Results**
In this work, the authors have established a proof of concept showing that data transfer between single-cell transcriptomic and body fluid proteomic modalities is possible. They identify specific gene expression and frequency changes on cell type level and use histopathologically staged data to infer early changes in lung fibrosis patients. Using machine learning, they demonstrate that protein signatures in lung lavage fluids correlating with lung function in a large cohort of lung fibrosis patients correspond with specific cell state and cell frequency changes in disease progression. Specifically, the study identifies an activated pericyte state that correlates with disease severity and shows that its presence is reflected by the complement regulatory factor CFHR1 in lung lavage fluid. The authors also discover that the de-differentiation of alveolar type-2 epithelial cells in lung fibrosis is reflected by lung lavage fluid and blood plasma levels of the protein CRTAC1, thus establishing a novel peripheral protein biomarker of the lung alveolar epithelial health status.

**Impact**
This work provides an integrated single-cell atlas of human lung fibrosis and establishes the correspondence of a set of peripheral protein biomarker signatures with cellular changes in the lung. The clinical utility of these novel biomarker signatures in monitoring disease progression can be addressed in prospective longitudinal follow-up studies. Conceptually, this study serves as important proof of concept for non-invasive cell state monitoring for future interceptive medicine.

Immunofluorescent images were recorded on an AxioImager.M2 microscope (Zeiss) using a Plan-Apochromat 20×/0.8 M27 objective.

## Immunohistochemical analysis of SSTR2 expression

Immunohistochemistry of SSTR2 on FFPE lung sections from ILD patients ($n = 53$) and controls ($n = 26$) was performed as previously described (Schniering *et al*, 2019a) using an automated single-staining procedure (Benchmark Ultra; Ventana Medical Systems) and a rabbit anti-SSTR2 antibody (abcam, ab134152, 1:50). Detection was finalized with respective secondary antibodies and the OptiView DAB Kit (Ventana Medical Systems). Images were acquired with the AxioScan.Z1 slidescanner (Zeiss) using a Plan-Apochromat 20×/0.8 M27 objective. For quantification of SSTR2 tissue expression, six randomly selected high power fields were taken per sample with the Zen 2.0 lite (blue edition) software and the percentages of positively stained pixels were automatically quantified using an in-house designed MATLAB script (MathWorks, MATLAB R2016b). For correlation of SSTR2 expression with the extent of lung fibrosis on the tissue level, the Ashcroft score was applied on adjacent picrosirius red-stained lung sections as previously described (Schniering *et al*, 2018, 2019b), and Pearson correlation analysis with SSTR2 tissue expression was performed.

# Data availability

Count tables of the Munich single-cell cohort as well as all custom analysis code can be accessed at https://github.com/theislab/2020_Mayr. Proteome raw data and MaxQuant processing tables can be downloaded from the PRIDE repository under the accession numbers PXD017145 (http://www.ebi.ac.uk/pride/archive/projects/PXD017145) (BALF) and PXD017210 (http://www.ebi.ac.uk/pride/archive/projects/PXD017210) (plasma).

**Expanded View** for this article is available online.

## Acknowledgements

We thank all the patients and their families for supporting the progress of science. We gratefully acknowledge the provision of human biomaterial and clinical data from the CPC-M bioArchive and its partners at the Asklepios Biobank Gauting, the Klinikum der Universität München, and the Ludwig-Maximilians-Universität München. We thank Ina Koch, Britta Peschel, and Annika Frank for managing the Asklepios Biobank and the CPC-M Bioarchive and supporting this study. We thank Igor Paron and Korbinian Mayr for expert support of the proteomics pipeline. We thank Sandy Lösecke and Elisabeth Graf for technical assistance in next-generation sequencing and Thomas Schwarz-mayr, Thomas Walzthöni, and Matthias Heinig for support with High-seq 4000 sequencing raw data and the single-cell processing pipeline. The Krt8/TROMA-I monoclonal antibody, developed by Brulet, P./Kemler, R., was obtained from the Developmental Studies Hybridoma Bank, created by the NICHD of the NIH and maintained at the University of Iowa, Department of Biology, Iowa City, IA, 52242. This work was supported by the German Center for Lung Research (DZL), the Helmholtz Association, the Max Planck Society, and the German Federal Ministry of Education and Research (BMBF), project Single-cell Genomics Network Germany, and the European Union's Horizon 2020 research and innovation program under grant agreement no. 874656 (discovAIR). This publication is part of the Human Cell Atlas (www.humancellatlas.org/publications/) and the LifeTime Initiative (https://lifetime-fetflagship.eu/). Open Access funding enabled and organized by Projekt DEAL.

## Author contributions

HBS conceived the research narrative, supervised the entire study, and wrote the paper. FJT, OE, MM, and JB supervised parts of the study. CHM, LMS, MA, and HBS performed single-cell data analysis and multi-modal data integration. CHM, LMS, GL, and HBS performed proteomics data analysis. GL analyzed clinical data. CHM and JS performed immunostainings. MS, IA, and PS collected single-cell data. CHM, PEG, and HBS collected proteomics data. NK, FR, ES, and ML collected clinical samples. SB, HA, BM, AH, AP, JB, MM, OE, FJT, and HBS provided resources. All authors read and approved the manuscript.

## Conflict of interest

The authors declare that they have no conflict of interest.

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
