## [Review Process File · EMBO Molecular Medicine]

Integrative analysis of cell state changes in lung fibrosis with peripheral protein biomarkers

Christoph Mayr, Lukas Simon, Gabriela Leuschner, Meshal Ansari, Janine Schniering, Philipp Geyer, Ilias Angelidis, Maximilian Strunz, Pawandeep Singh, Nikolaus Kneidinger, Frank Reichenberger, Edith Silbernagel, Stephan Boehm, Heiko Adler, Britta Maurer, Anne Hilgendorff, Michael Lindner, Antje Prasse, Jürgen Behr, Matthias Mann, Oliver Eickelberg, Fabian Theis, and Herbert Schiller
DOI: 10.15252/emmm.202012871

Corresponding authors: Herbert Schiller (herbert.schiller@helmholtz-muenchen.de)

Review Timeline:

Submission Date:	3rd Jun 20
Editorial Decision:	7th Jul 20
Revision Received:	9th Nov 20
Editorial Decision:	7th Dec 20
Revision Received:	5th Jan 21
Accepted:	19th Jan 21

Editor: Jingyi Hou

Transaction Report:

7th Jul 2020

Dear Dr. Schiller,

Thank you for the submission of your manuscript to EMBO Molecular Medicine. We have now received feedback from the two of the three referees whom we asked to evaluate your manuscript. One of the referees who initially accepted to review the manuscript finally dropped out. As you will see from the reports below, the other two referees acknowledge the potential interest of the study. However, they also raise substantial concerns about your work, which should be convincingly addressed in a major revision of the present manuscript.

I think that the referees' recommendations are rather clear and there is no need to reiterate their comments. In particular, the clarity in data/study presentation needs to be improved to make the findings easily accessible to the general audience of EMBO Molecular Medicine (Both referees provide constructive suggestions in this regard). Moreover, attention should be given to placing the findings in the context of previous literature and to highlighting the novelty of the current study.

All other issues raised by the referees need to be satisfactorily addressed as well. We would welcome the submission of a revised version within three months for further consideration. Please note that EMBO Molecular Medicine strongly supports a single round of revision and that, as acceptance or rejection of the manuscript will depend on another round of review, your responses should be as complete as possible.

We are aware that many laboratories cannot function at full efficiency during the current COVID-19/SARS-CoV-2 pandemic and have therefore extended our "scooping protection policy" to cover the period required for a full revision to address the experimental issues. Please let me know should you need additional time, and also if you see a paper with related content published elsewhere.

I look forward to receiving your revised manuscript.

Yours sincerely,

Jingyi Hou

Jingyi Hou
Editor

*** Instructions to submit your revised manuscript ***

** PLEASE NOTE ** As part of the EMBO Publications transparent editorial process initiative (see our Editorial at <https://www.embopress.org/doi/pdf/10.1002/emmm.201000094>), EMBO Molecular Medicine will publish online a Review Process File to accompany accepted manuscripts.

To submit your manuscript, please follow this link:

Link Not Available

- 1) a .docx formatted version of the manuscript text (including Figure legends and tables). Please make sure that the changes are highlighted to be clearly visible to referees and editors alike.
- 2) separate figure files*
- 3) supplemental information as Expanded View and/or Appendix. Please carefully check the authors guidelines for formatting Expanded view and Appendix figures and tables at <https://www.embopress.org/page/journal/17574684/authorguide#expandedview>
- 4) a letter INCLUDING the reviewers' reports and your detailed responses to their comments (as Word file)

Also, and to save some time should your paper be accepted, please read below for additional information regarding some features of our research articles:

5) The paper explained: EMBO Molecular Medicine articles are accompanied by a summary of the articles to emphasize the major findings in the paper and their medical implications for the non-specialist reader. Please provide a draft summary of your article highlighting

6) For more information: There is space at the end of each article to list relevant web links for further consultation by our readers. Could you identify some relevant ones and provide such information as well? Some examples are patient associations, relevant databases, OMIM/proteins/genes links, author's websites, etc...

7) Author contributions: the contribution of every author must be detailed in a separate section (before the acknowledgments).

8) EMBO Molecular Medicine now requires a complete author checklist (<https://www.embopress.org/page/journal/17574684/authorguide>) to be submitted with all revised manuscripts. Please use the checklist as a guideline for the sort of information we need WITHIN the manuscript as well as in the checklist. This is particularly important for animal reporting, antibody dilutions (missing) and exact p-values and n that should be indicated instead of a range.

9) Every published paper now includes a 'Synopsis' to further enhance discoverability. Synopses are displayed on the journal webpage and are freely accessible to all readers. They include a short stand first (maximum of 300 characters, including space) as well as 2-5 one sentence bullet points that summarise the paper. Please write the bullet points to summarise the key NEW findings. They should be designed to be complementary to the abstract - i.e. not repeat the same text. We encourage inclusion of key acronyms and quantitative information (maximum of 30 words / bullet point). Please use the passive voice. Please attach these in a separate file or send them by email, we will incorporate them accordingly.

You are also welcome to suggest a striking image or visual abstract to illustrate your article. If you do please provide a jpeg file 550 px-wide x 400-px high.

10) A Conflict of Interest statement should be provided in the main text

11) Please note that we now mandate that all corresponding authors list an ORCID digital identifier. This takes <90 seconds to complete. We encourage all authors to supply an ORCID identifier, which will be linked to their name for unambiguous name identification.

Currently, our records indicate that the ORCID for your account is 0000-0003-1914-3215.

Link Not Available

12) The system will prompt you to fill in your funding and payment information. This will allow Wiley to send you a quote for the article processing charge (APC) in case of acceptance. This quote takes into account any reduction or fee waivers that you may be eligible for. Authors do not need to pay any fees before their manuscript is accepted and transferred to our publisher.

Each figure should be given in a separate file and should have the following resolution:
Graphs 800-1,200 DPI

Photos 400-800 DPI

*Additional important information regarding figures and illustrations can be found at <http://bit.ly/EMBOPressFigurePreparationGuideline>

***** Reviewer's comments *****

Referee #1 (Comments on Novelty/Model System for Author):

Not relevant

Referee #1 (Remarks for Author):

Mayr et al describe a study of lung fibrosis, in which they integrate single-cell RNAseq with proteomic analysis of body fluids (i.e. BALF and serum). They analyzed several cohorts by scRNAseq to associate between clinical features and the distribution of cell populations. Next, they performed proteomic analysis of the proximal BAL fluid to associate the proteins with the clinical state. Next, they integrate the two data layers to associate between the secreted proteins and the cells of origin. Finally, they added serum proteomics to identify blood biomarkers of the clinical state. Overall, this is a very interesting study, which shows the power of integration of the scRNAseq data, bulk data and secreted proteomes. All of these analyses were performed using state of the art technologies and may lead to important clinical insights.

The manuscript is also nicely written and presented. However, it seems like writing improvement could be useful to broaden the readership of the paper. On one hand, the manuscript is largely bioinformatic, and would be of interest to the proteogenomic community, however the clinical aspects and the large number of abbreviations make it difficult to follow. On the other hand, the manuscript is expected to be of high interest to the specific clinical community, but the lack of validation and proper description of the biological results reduce the applicability to this community. Specific comments:

1. UMAPs of scRNAseq show nicely the separation of cell types, and separation between disease states (figures 1,2). Beyond these, it is also important to see the variance between patients. Are there differences between the cell populations among patients with the same clinical diagnosis? A reference to this variance is critical to assess the relevance of the results to the larger patient population.
2. The scRNAseq sections lack sufficient description of the biological results and their relevance to the clinical feature. What are the clinical implications of the changes in cell populations and the implications of the changes in specific proteins?
3. The authors should discuss the differences in cell populations vs. the differences in gene expression within each cell population. These biological aspects are missing altogether and would be very important to add to the biological significance of their study.
4. The authors claim that the BALF is difficult to analyze due to the high dynamic range. However this statement needs to be shown. Since this is not a well-studied body fluid, a general description of the identified proteins, dynamic range and comparison to serum is important to the proteomic

readership of the paper.

5. The importance and focus of CRTAC1 is not convincing and the reference to it throughout the paper is confusing. First of all, as a marker, a protein that is downregulated in disease state is always less preferable than an upregulated biomarkers. Second, the cell association (figure 4g) seems rather weak. Third, the general association with lung function, rather than association with specific pathologies reduces its value as a molecular biomarker. Last, in terms of manuscript organization, it is rather confusing to read about the same protein in every section, especially given that the functional and data justification is not very strong every time. It would be clearer if this protein is discussed at the end, and only then the various data layers of this protein are shown.

6. The papers should include histological validation (IHC/IF) of protein expression in relation to the scRNAseq results. The leap from scRNAseq to secreted protein is very large, and requires additional analyses of the proteins in the tissue context, with respect to the cell types identified by scRNAseq.

Minor comments:

1. The authors should add a brief mention of the scRNA process in the results. It is not mentioned at all.
2. Figure 1a-b: The authors indicate that cell populations found mainly in patients were called 'activated'; however, AM activated doesn't seem to be in the patients, but higher levels are seen in the controls.
3. I suggest to change the color code of figure S1b. A clearer separation of the colors of the controls vs. the disease would help interpreting the results (rather than comparing to the main figures).
4. The integration of scRNAseq of three cohorts mentions 63 patients, however the specific numbers per cohort (patient/control) don't add up. Should be revised.
5. The sentence "Gene expression changes in disease were more similar within the epithelial, mesenchymal and leukocyte lineages (Fig. 2d)." (p4) is not clear. What do the authors mean by changes being more similar? Should be revised.
6. Integration with bulk RNAseq should be more detailed. How many patients were analyzed? What are the biological insights from this integration?
7. Reference to figure 2e does not seem to be in the right place.
8. The distinction between ELF and tissue leakage proteins is not clear. I presume the authors mean that ELF proteins (as they defined them) are expected to have specific functions in the fluid, as opposed to randomly secreted proteins. However, epithelial lining fluid also includes the tissue leakage proteins and it is not known whether they may also have extracellular function in the fluid. Their selection of proteins to focus on should be better explained.
9. Figure S3d should indicate what the color scale refers to.
10. Method section of the proteomic analysis indicates a non-linear gradient of water-acetonitrile. The details of this gradient should be indicated especially since these are non-standard running conditions. Normally, gradients do not reach 60% buffer B.
11. What are the details of the trypsin digestion? Usually runs are longer than 1h (e.g. ON). Which trypsin was used here?
12. Biognosys section of DIA data analysis should move to the next section in the methods rather than the MS part.

Referee #3 (Comments on Novelty/Model System for Author):

The data presented in this manuscript are well-analyzed and the overall utility is high. Some of the transcriptomic data are from published datasets, limiting the novelty, but the new analysis and addition of an orthogonal proteomic dataset enriches these datasets. The authors could add some

detail clarifying what was learned from combining multiple transcriptomic datasets on beyond what is already known. The potential for medical impact is high as this is a devastating and poorly understood disease, but it would be nice to have further clarity as to how this approach would be for identifying subclinical biomarkers prior to lung function decline.

Referee #3 (Remarks for Author):

In this manuscript, Mayr/Simon et al. describe an integrated computational analysis combining single cell transcriptomic and bulk proteomic measures taken from donors with pulmonary fibrosis and matched controls. By combining their own datasets with previously published analyses, they generate a large single cell dataset of 63 pulmonary fibrosis patients. In parallel, they measure biomarkers in lung lavage and plasma from a separate patient cohort. By developing new computational methods to analyze these datasets in tandem, they identify the type 2 cell protein CRTAC1 as a biomarker of lung function decline in pulmonary fibrosis.

Overall, this effort represents an important compilation and new analysis of novel and publicly available datasets that will serve as a valuable resource for the pulmonary fibrosis field. The motivation for the study is clear and obviously important. In particular, the idea of using proteomic biomarkers to predict cell state changes in difficult-to-access tissues is exciting, and this approach is a potential proof-of-principle for this concept. With this in mind, some aspects of the current analysis make the interpretation difficult to follow, and the full implications of this work are not entirely clear. In particular, a major challenge in presenting large datasets is in clearly outlining the rationale behind the selected analyses and how the authors reached a given conclusion. While it is of course impossible to follow the data down every possible path, it is important for the reader to understand the rationale behind each decision and interpretation and for the analysis to be well-grounded in prior efforts. Addressing the major points below may help the authors to clarify their findings and the potential impact for the field:

Major Points

1. Cell type assignments and identification of disease-associated changes

The authors present an in-depth analysis of single cell data gathered across several patient cohorts from three sites. Overall, the integration of these datasets is well done, and in particular the comparisons of changes in from healthy to disease states across these datasets is useful for identifying genes important to the pathogenesis of IPF.

However, throughout the manuscript it is difficult to keep track of the changes in cellular identity occurring with disease onset and progression. Further clarification of the results might illuminate their interpretations and conclusions more fully. In particular, they seem to not always distinguish between changes in the number of a certain cell type (for example, type 2 cells are known to be lost in pulmonary fibrosis) or in the identity of a certain cell type (are there genes expressed in IPF type 2 cells that are not expressed in healthy controls?). Throughout the manuscript, this makes it difficult to follow their interpretations and to ground this new analysis with what is already known from prior studies about the etiology of this disease.

- A major advance in this paper is the careful integration of multiple datasets from multiple studies. As there are many published datasets on both healthy and IPF human lung (including the papers describing the datasets used in this paper), it would be helpful for the authors to ground their analysis more substantially in this prior work. In particular, do they learn anything new from this analysis that was not observed in these prior studies?

- In Figure 1, they choose to subset the data based on one marker gene per population, but do not comment on whether this marker gene changes in disease. For example, they choose to look at EPCAM+ epithelial cells, but EMT (and associated loss of EPCAM) in epithelia is associated with pulmonary fibrosis - is it possible they are missing some unique epithelial populations (or intermediate "EMT" populations) in patient samples using this approach? Are there cells that don't fall into these categories?
- They mention the assignment of "activated cell states" that are associated with IPF, but nowhere in the manuscript are these states enumerated and characterized. It seems like the analysis they performed should include all of these data; it would be nice to have them included in the manuscript.
- Related to this, in some analyses it is difficult to understand whether they are measuring changes in cell identity/gene expression or the number of a certain cell type. For example, is Figure 1j showing that ciliated cells are increased in later stage pulmonary fibrosis, or that there are certain genes increased within the ciliated cell population in pulmonary fibrosis. If it is the former, what are the genes that are changing in this population?
- In Figure 2., the use of available data to track disease progression and map samples to early disease states is nice, but it isn't clear what the actual interpretation of this analysis is. Is the myofibroblast signature upregulated in the early stages of disease because fibrotic infiltration (and an increase in fibroblast gene expression in the tissue) is an early event, or because of early transformation of existing fibroblasts towards a more disease-like myofibroblast state? After clarifying which of these hypotheses they intend to propose, can they provide further evidence (even in the form of a clearer analysis of this data) to support this claim?
- Overall, it would really clarify their findings if they were able to more clearly relate the observed data to changes in cell number and identity in pulmonary fibrosis, in particular observations known from prior work (for example, do they see evidence of bronchiolization, infiltration/transformation of macrophages, loss of type 2 cells) and from the analysis of the datasets they use in this paper from prior single cell studies. Again, this is likely just a matter of including the relevant graphs in the supplemental data or making an interactive data browser available.

2. Analysis of proteomic datasets and comparison to single cell sequencing results

The authors present a new dataset of proteomic analysis of lavage fluid and plasma from patients with pulmonary fibrosis. From these data, they validate known biomarkers and propose new signatures correlated with clinical features and disease progression. Finally, they map between this dataset and their single cell transcriptomic analysis to test the utility of lavage/plasma biomarkers for predicting cell state changes in disease-affected tissue. While biomarker studies have been done in IPF, this is a large dataset and represents a first effort to compare with sequencing dataset. The data and analysis approach are of high-quality and of clear use to the field. However, further clarification on a few points would help support the author's interpretation of these result. In particular, as in Major Point (1), it is occasionally difficult to follow the rationale for the selected analysis approach and the consequences for the subsequent interpretation of these results:

- The authors choose to focus their analysis on secreted proteins and limit their interpretation of proteins released as a likely consequence of tissue damage. It is not clear why they make this decision, as protein accumulation in the lung lining fluid and in plasma is a clear marker of tissue damage associated with disease. Can the authors clarify the motivation behind this decision, and

how this might change their interpretation of the results?

- Much of the data validates known features of pulmonary fibrosis, such as an increase in immune infiltration in the lung associated with measurement of cytokine and other pro-inflammatory proteins. It is not clear from the presentation of these data what exactly the proteomic data are showing beyond this finding. For example, are the authors able to distinguish increased immune cell infiltration versus a change in protein composition or cytokine secretion from these cells and therefore make a prediction about potential mechanistic drivers of disease?
- The analysis of CRTAC1 as a biomarker is nicely done and it is clear that this gene is downregulated in IPF (and consequently the loss of this protein can be readily measured as a biomarker). It seems that CRTAC1 is downregulated in type II cells in disease and that type II cells are lost in disease progression. It seems clear that there should be an association between loss of type II cells and loss of lung function. Do they see a similar response in other genes/proteins associated with type II cells, or is there something unique about this gene beyond its expression in this cell type? If the latter, can the authors speculate as to the importance of this gene in disease?
- The idea of developing a biomarker to measure lung function decline is exciting, but its clinical utility relies on the ability to detect lung function decline at a sub-clinical level (e.g. in early stages of disease), as it is relatively simple to measure clinically. The analysis of relative contribution of cell types to the biomarker signature seems really useful and promising for subclinical phenotyping. It would be useful to further understand how they might also detect subclinical or more subtle early changes, what the sensitivity of this approach might be, and how these data could be analyzed and interpreted. Can the authors comment further on how this might be accomplished using this approach?
- Related to this, in Figure 2i-j they measure gene expression changes predicted to map to early stages of disease. Can they use this result together with their later machine learning analysis to predict protein biomarker changes that might occur early in disease?

Minor Points

- Early in the manuscript, the authors discuss in detail the effort of the Human Cell Atlas project. Why the focus on this particular effort - are all of the datasets used in this study funded by this group?
- The use of bbknn for batch correction seems to work nicely, but this method relies on a relatively one-to-one mapping of states across the selected "batch" variable. As the authors later use a regression analysis to look into demographic covariates, it seems like they don't necessarily expect these datasets to be well-matched. Would it be possible to use the approach used in Fig S2 to also perform batch correction for Fig. 1?
- In the introduction they mention the clinical utility of biomarkers to measure patient heterogeneity in a complex disease like IPF. They later discuss this in the context of acute exacerbations, but it might also be interesting to learn if they see any evidence for different etiologies or subtypes of pulmonary fibrosis in different patients. Are they able to understand anything further about this from these data?
- The patient cohorts used for the proteomic versus transcriptomic analysis are non-overlapping. The authors comment that this is a major limitation of this approach. Can they further clarify the differences between these cohorts and what the specific limitations are for the interpretation of the

results? Do they expect to see differences in any particular parameters? Do they have paired transcriptomic/proteomic datasets for any individual to serve as validation for their approach?

- The authors see the strongest association in their proteomic dataset with changes in epithelial lineages, possibly driven by their focus on secreted proteins. The epithelium is highly secretory and responsive to damage and environmental changes. Is this the likely reason for this strong association, or are there other possible interpretations?

Point by point reply to the reviewers comments

Referee #1 (Comments on Novelty/Model System for Author):

Not relevant

Referee #1 (Remarks for Author):

Mayr et al describe a study of lung fibrosis, in which they integrate single-cell RNAseq with proteomic analysis of body fluids (i.e. BALF and serum). They analyzed several cohorts by scRNAseq to associate between clinical features and the distribution of cell populations. Next, they performed proteomic analysis of the proximal BAL fluid to associate the proteins with the clinical state. Next, they integrate the two data layers to associate between the secreted proteins and the cells of origin. Finally, they added serum proteomics to identify blood biomarkers of the clinical state. Overall, this is a very interesting study, which shows the power of integration of the scRNAseq data, bulk data and secreted proteomes. All of these analyses were performed using state of the art technologies and may lead to important clinical insights.

The manuscript is also nicely written and presented. However, it seems like writing improvement could be useful to broaden the readership of the paper. On one hand, the manuscript is largely bioinformatic, and would be of interest to the proteogenomic community, however the clinical aspects and the large number of abbreviations make it difficult to follow. On the other hand, the manuscript is expected to be of high interest to the specific clinical community, but the lack of validation and proper description of the biological results reduce the applicability to this community.

We thank the reviewer for the constructive and positive evaluation of our work. Please find a detailed response below.

Specific comments:

1. UMAPs of scRNAseq show nicely the separation of cell types, and separation between disease states (figures 1,2). Beyond these, it is also important to see the variance between patients. Are there differences between the cell populations among patients with the same clinical diagnosis? A reference to this variance is critical to assess the relevance of the results to the larger patient population.

We carefully analyzed variance of cell type and state frequencies between individual patients and disease groups. This analysis is now shown in Figure 3 and Figure S2 of the revised manuscript:

Results page 6:

Differences in cell type frequencies between individual samples can be caused by true biological differences, as well as differences in cell isolation protocols and scRNAseq platforms used. Indeed, we observed large variance in cell population frequencies across cohorts, and disease conditions (Fig. S2c). After performing dimension reduction using PCA, we observed larger variation in principal components one and two among ILD patients compared to control samples, indicating increased heterogeneity in disease (Fig. 3a). Principal component two separated samples obtained from control donors and ILD patients across all three cohorts (Fig. 3b). This observation motivated us to ask if cell type frequencies alone could distinguish ILD samples from controls. Therefore, we trained a random forest model based on the cell type proportions to predict disease status. This model achieved a mean accuracy of 83% derived from 5 fold cross-validation (Fig. 3c), demonstrating that the cell type frequencies showed robust differences in disease. Most important for the models prediction accuracy were changes in frequency of disease induced cell states as well as several parenchymal cell identities (Fig. 3d). As expected, the decrease of AT1 and AT2 cells was important for the predictions. Interestingly, the top importance score in the model was achieved by the recently discovered aberrant basaloid cells^{28,29}, suggesting that this cell state is indeed very disease specific (Fig. 3d).

Figure S2. Differential detection and cell type frequency analyses.(c) Boxplots illustrate the frequencies (x-axis) of cell types (y-axis) across samples from ILD patients (yellow) and control donors (purple) from the three indicated cohorts.

To leverage the power of bulk RNA-seq data archived in public databases, we used our ILD single cell atlas to determine possible cell type frequency changes in such datasets. A recent study used quantitative microCT imaging and tissue histology on biopsies to stratify lung tissue of idiopathic pulmonary fibrosis (IPF) patients into different stages (IPF stage 1-3) marked by increasing extent of fibrotic remodeling (lower alveolar surface density and higher collagen content)²². Thus, the RNA-seq profiles of these staged patient samples presumably depict disease progression within patients. We calculated enrichment of our cell state signatures across the three stages of IPF progression and observed significant changes of many cell types already in early stage IPF-1, **which still harbors more alveolar cell identities compared to the more advanced stages IPF-2 and IPF-3** (Fig. 3e). This included

the myofibroblast signature (Fig. 3f) that was clearly upregulated early in progression as well as the aberrant basaloid cells. Other cell signatures, such as the plasma cells, showed a gradual increase from IPF1 to IPF3, while for instance the increase in ciliated cell frequency was observed only from IPF stage 2 onwards (Fig. 3f). Increases in airway cell frequencies with advanced stages are likely the consequence of the well described ‘bronchiolization’ of the distal lung with metaplastic epithelial cells in IPF. Importantly, our analysis identifies cell state shifts and cell frequency changes that precede this bronchiolization. The appearance of the aberrant basaloid cells together with activated fibroblast states peaked already at the IPF-1 stage, indicating that these represent early events in disease progression. Notably, also several immune cell types, including B- and T- lymphocytes were increased already in IPF-1.

Figure 3. Disease progression alters cell type frequencies. (a) Principal components one (x-axis) and two (y-axis) illustrate the cell type frequencies in reduced dimensions. The shape of each point corresponds to the study cohort and the points are colored by disease phenotype. (b) Boxplots display principal component two across ILD patients (orange) and control samples (purple) for the three study cohorts. (c) Boxplot depicts random forest model prediction accuracies derived from five fold cross-validation using the original and permuted labels. (d) Barplot shows the random forest importance scores for the top ten most informative features. (e) The heatmap shows changes of the indicated cell type signatures in published bulk RNA-seq data (GEO GSE124685) across different histopathological stages that represent increasing extent of fibrosis from stage 1-3, as determined by quantitative micro-CT imaging and tissue histology²². Samples used in this study were 10 IPF and 6 control patients. (f) The heatmaps show z-scores for the individual marker genes of the indicated cell types across IPF stages and controls.

2. The scRNAseq sections lack sufficient description of the biological results and their relevance to the clinical feature. What are the clinical implications of the changes in cell populations and the implications of the changes in specific proteins?

We strengthened our manuscript with more details and discussion on biological results throughout the entire paper in essentially all chapters. For example, we have now discussed the potential pathophysiological relevance of observed cell state and protein changes (e.g. complement regulation via CFHR1 in activated pericyte state). In particular, the activated pericyte state described in Figure 7 of the revised manuscript is novel and we show that these cells are significantly associated with the degree of fibrotic remodeling and potentially involved in immune cell recruitment via their secretion of chemokines and complement regulation through CFHR1. Furthermore, in Figure 8 we have expanded our analysis on the relevance of CRTAC1 in reporting the de-differentiation of AT2 cells towards the novel aberrant basaloid cell state in lung fibrosis. This is very strong first evidence for a novel biomarker of AT2 cell health, which might have important use cases for future prospective clinical studies.

See discussion page 20:

In idiopathic pulmonary fibrosis (IPF), the most common form of lung fibrosis, the progressive replacement of lung parenchyma with scar tissue leads to respiratory failure with a median survival time of 2-4 years after diagnosis. Current models of disease pathogenesis propose that a combination of repetitive (micro)injuries to susceptible alveolar epithelial cells (AEC) with an aberrant repair response causes pathological interactions of AEC with fibroblasts and subsequent accumulation of scar tissue⁵⁵. Human genetic data and preclinical models show that epithelial injury possibly caused by environmental exposures, can drive subsequent fibrosis, with a combination of genetic predisposition and aging thought to be related to the failed regenerative response in IPF. **The recent discovery of a transitional stem cell state that transiently appears in normal lung regeneration but persistently accumulates in lung fibrosis enables a new perspective on pathogenesis of IPF^{30,38,56}. This transitional stem cell state in mice is highly similar to the aberrant basaloid cells in IPF^{28,29} and features the expression of several pro-fibrogenic factors that may activate mesenchymal cells. We show that the main source of CRTAC1 in the human body is the alveolar AT2 cell, and that CRTAC1 is downregulated early in a de-differentiation trajectory of AT2 cells towards the aberrant basaloid state. Our data suggests that this cellular transition is reflected in plasma proteomes by declining abundance of CRTAC1. We propose that CRTAC1 protein levels in plasma and lavage fluids specifically report the AT2 cell health status. This novel biomarker is thus a promising candidate for future prospective trials in various settings, including the monitoring the degree of distal lung involvement during virus induced pneumonia as currently seen in the COVID-19 pandemic. Supporting our hypothesis, a recent preprint illustrates that CRTAC1 is downregulated in plasma of hospitalized COVID-19 patients compared to SARS-CoV-2 negative controls⁵⁷.**

Perivascular cells have been shown to be key contributors to organ fibrosis, including the lung⁵⁸, and a pericyte to myofibroblast transition in lung fibrosis has been proposed⁵⁹. As our single cell analysis demonstrates, the previous functional studies of PDGFRB+ cells isolated from lungs underappreciated the heterogeneity of PDGFRB+ cells, which not only contain perivascular pericytes but also fibroblast populations and smooth muscle cells. In this work we discover a highly disease-specific SSTR2+/CFHR1+ pericyte state that features a pro-inflammatory phenotype with expression of various chemokines. Interestingly, the expression of the potent complement regulatory factor CFHR1 in this cell state may explain the previously observed deregulation of complement in IPF.

Further functional investigations on this novel pericyte state in lung fibrosis will shed light on its potential direct relevance in disease progression. Interestingly, visualizing radiolabelled somatostatin analogues targeting the *SSTR2* receptor have recently been proposed for the visualisation of fibrotic changes in ILD^{43,60,61}.

3. The authors should discuss the differences in cell populations vs. the differences in gene expression within each cell population. These biological aspects are missing altogether and would be very important to add to the biological significance of their study.

Thank you. That was an important point and we believe the revisions of this aspect really strengthened the manuscript. We added additional analysis contrasting inferred changes in cell type frequency with changes in gene expression to the revised manuscript. We have ranked the cell types based on their contribution to the BALF proteome signatures (Fig. 5e). The protein signature is a composite representation of both cell type frequency and gene expression changes. Single cell analysis enables us to disentangle both layers of regulation.

Results section page 12:

Next, we aimed to understand the relationship between cell type specific transcriptional changes and distinct signatures in the bronchoalveolar lavage fluid. To accomplish this aim, we integrated the results from two multivariate regression analyses: 1) protein associations with meta lung function and 2) cell type specific RNA associations with ILD status. Correlation of the t-values derived from both regressions revealed that the greatest correspondence between protein signatures associated with meta lung function and expression changes occurred in club and basal cells as well as the alveolar epithelium (Fig. 5e). However, the measured bulk protein profiles in BALF can be affected by two types of alterations: 1) changes in cell type frequency and 2) changes in gene expression. Therefore, we inferred cell type frequency changes by performing deconvolution analysis on a large bulk expression data set containing ILD patients and healthy controls from the Lung Tissue Research Consortium (LTRC; GSE47460). The top regulated BALF proteins were often altered both on gene expression and cell type frequency levels (Fig. 5f), which importantly can be distinguished using

scRNASeq data.

Figure 5... (f) The heatmap illustrates gene expression changes associated with lung fibrosis across indicated cell types (columns) for selected BALF protein biomarkers (rows). The dotplot visualizes the frequency changes of the indicated cell types inferred from the deconvolution of bulk mRNA data of ILD samples compared to controls (GSE47460). Samples used in this study from the LTRC n= 254 ILD patients and n= 108 controls.

4. The authors claim that the BALF is difficult to analyze due to the high dynamic range. However this statement needs to be shown. Since this is not a well-studied body fluid, a general description of the identified proteins, dynamic range and comparison to serum is important to the proteomic readership of the paper.

Following up on this comment we systematically compared the BALF and plasma MS data to illustrate the high dynamic range and loss of sensitivity in BALF MS due to plasma protein leakage. We found a strong anti-correlation of plasma protein abundance with sensitivity of MS analysis (Fig. S3a). Many proteins are detected both in plasma and BALF - within these proteins we find proteins that are having a higher abundance in the BALF. These proteins, including CRTAC1 and known cell type markers such as BPIFB1 expressed by airway secretory cells and AT2 specific SFTPB, are thus produced locally and transpire to the plasma.

Results section page 8:

‘From 124 patients (95 ILD and 29 non-ILD) that passed quality control criteria, we quantified a median number of 835 proteins per individual patient, resulting in a total of 1513 unique proteins that were quantified in at least 20 patients (Fig. 4b and Table S6). This is a very good depth of analysis given that BALF is difficult to analyze by mass spectrometry because of the high dynamic range of protein copy numbers present due to plasma protein leakage. To illustrate a correlation of the extent of plasma protein leakage with the depth of proteome analysis in our BALF samples, we quantified the mass fraction of the top 100 abundant proteins in patient plasma in the BALF proteomes. The number of detected proteins in BALF inversely correlated with the relative proportion of plasma proteins present (Fig. S3a), however this was not strongly associated with plasma LDH, which serves as a marker for tissue damage in patient blood. Using a quantitative comparison of proteins detected in both the ILD patient plasma and BALF proteomes revealed proteins that are detected in plasma proteomes but are quantitatively enriched in BALF, suggesting they are produced locally in the lung and then transpire to the plasma (Fig. S3b).’

Figure S3. Defining the epithelial lining fluid proteome in a large interstitial lung disease patient cohort. (a) The fraction of plasma proteins (defined by top 100 abundance rank in plasma) is shown as a percentage of total mass fraction of all identified proteins in BALF. Note that this fraction negatively correlates with the total number of identified proteins in BALF. The colors represent plasma LDH values of individual patients. (b) The scatter plot shows mean MS-intensities of proteins identified in both plasma and BALF from all ILD patients. The red circle indicates locally enriched proteins that are more abundant in the lung compared to plasma samples.

5. The importance and focus of *CRTAC1* is not convincing and the reference to it throughout the paper is confusing. First of all, as a marker, a protein that is downregulated in disease state is always less preferable than an upregulated biomarkers. Second, the cell association (figure 4g) seems rather weak. Third, the general association with lung function, rather than association with specific pathologies reduces its value as a molecular biomarker. Last, in terms of manuscript organization, it is rather confusing to read about the same protein in every section, especially given that the functional and data justification is not very strong every time. It would be clearer if this protein is discussed at the end, and only then the various data layers of this protein are shown.

We put the *CRTAC1* data to the end of the manuscript (now described in Figure 8) as suggested. Furthermore, we substantially extended our analysis and discussion on the biological relevance of this marker. The specific association of *CRTAC1* to AT2 cells is very strong as shown in Figure 7b of the revised manuscript. The downregulation of *CRTAC1* marks the beginning of an AT2 de-differentiation trajectory towards the novel aberrant basaloid state, thus reporting a very specific cell state change with high clinical relevance.

Results section page 16:

***CRTAC1* is a novel peripheral protein biomarker of AT2 cell health status in the lung**

The BALF protein with the highest and most significant positive association in our multivariate regression meta lung function analysis was the cartilage acidic protein 1 (*CRTAC1*), whose function in the lung is currently unknown (Fig. 4e, Fig. 8a). Our single cell atlas revealed specific expression of *CRTAC1* in lung lymphatic endothelium, airway club cells, and most prominently in alveolar type-2 epithelial (AT2) cells (Fig. 8b, Fig. S7b). On the whole body level the mRNA expression of *CRTAC1* was highest in the lung (Fig. 8c). Expression of *CRTAC1* in alveolar epithelial cells was consistently downregulated in ILD samples compared to controls in all three patient cohorts analyzed by single cell RNA-seq (Fig. 8d). Also re-analysis of published bulk transcriptomes confirmed a highly significant downregulation of *CRTAC1* mRNA in the lung of ILD patients compared to healthy controls and COPD patients (Fig. 8e).

To identify gene programs within AT2 cells that are associated with *CRTAC1* expression, we performed gene-gene correlation analysis within SFTPC+ AT2 cells. Positively and negatively correlated genes were identified and those correlations were reproducible across all three cohorts (Fig. 8f). Genes positively correlated with *CRTAC1* across all cohorts were significantly enriched for categories that are consistent with a normal AT2 cell identity, including surfactant genes, and cholesterol biosynthesis genes. The anti-correlated genes were enriched for immune and inflammatory processes, as well as extracellular matrix and intermediate filament genes (Fig. 8g).

Analysis of transcriptional regulators revealed that the expression of the transcriptional activator C/EBP-delta (*CEBPD*) was highly correlated with *CRTAC1* expression (Fig. 8h). *CEBPD* expression and activity is induced by glucocorticoids and has a role in AT2 cell differentiation during lung development^{45,46}. Interestingly, the levels of *CRTAC1* in isolated human AT2 cells increase upon differentiation with glucocorticoids⁴⁷, which are known to be essential for alveolar maturation in lung development⁴⁸. Thus, we speculate that the downregulation of *CRTAC1* in AT2 cells of ILD patients may hint at currently uncharacterized changes in glucocorticoid signaling in these cells. We also performed upstream regulator analysis in Ingenuity Pathway Analysis (IPA) to predict the activity of transcriptional regulators based on the correlated or anti-correlated gene profiles (Fig. 8i). This analysis identified *ETV5* as a potential regulator of the *CRTAC1* correlated gene program in AT2. This is in line with the fact that *ETV5* has been shown to be essential for AT2 cell maintenance in vivo, as deletion of *Etv5* from mouse AT2 cells produced gene and protein signatures characteristic of differentiated alveolar type I (AT1) cells⁴⁹. Thus, our finding here is consistent with the notion that *CRTAC1* is associated with a normal healthy and highly differentiated AT2 cell state.

Figure 8. CRTAC1 protein abundance in BALF and plasma proteomes reports AT2 cell health. (a) The scatter plots shows the positive correlation of CFHR1 in BALF (MS-intensity, x-axis) with meta lung function (y-axis). (b) UMAP visualizes embedding of single cells colored by gene expression for CRTAC1, which is specifically expressed in alveolar type-2 (AT2), Club and lymphatic endothelial (Lymp_EC) cells. (c) Relative expression level of CRTAC1 across human organs. (d) The box plots illustrate differences in mRNA detection for CRTAC1 in alveolar epithelial cells from fibrosis patients compared to control samples across the three patient cohorts. (e) Relative gene expression levels of CRTAC1 in GSE47460. Dots represent average expression in the

tissue of individual patients. The line represents the mean. *CRTAC1* is significantly downregulated in ILD but not COPD patients (One-way ANOVA). (f) For each single cell cohort, the gene-gene correlations with *CRTAC1* within the SFTPC+ AT-2 cells were calculated. The indicated genes were selected based on their common direction of correlation across cohorts. (g) The bar graph shows the gene categories most strongly correlated with *CRTAC1* based on “Uniprot keywords”. (h) The bar graph shows the gene categories most strongly correlated with *CRTAC1* belonging to the GO category of “transcription regulators”. (i) The bar graph shows the top correlated transcriptional regulators, predicted by ingenuity Pathway Analysis (IPA) for the *CRTAC1* gene-gene correlations. (j) Diffusion map of human AT2 cells colored by cell type identity and inferred pseudotime. (k) The line plot illustrates smoothed expression levels of the indicated genes across the human AT2 pseudotime trajectory. (l) Diffusion map of mouse AT2 cells colored by cell type identity and inferred pseudotime. (m) The line plot illustrates smoothed expression levels of the indicated genes across the mouse AT2 pseudotime trajectory. (n) Immunofluorescence analysis of *SPRR1A*, *KRT8* as well as SFTPC in IPF (n=3) and control samples (n=2). (o) A high throughput experimental workflow for plasma proteomics⁵⁰ allowed for profiling of two independent cohorts of ILD patients (Munich, n=30 and Hannover, n=81; healthy age matched controls, n=30). All proteins quantified in plasma are shown, ranked by their abundance measured by mass spectrometry (MS-intensity). (p) The indicated proteins from the plasma analysis were selected based on their common direction of correlation with patient lung function in two independent patient cohorts with distinct clinical characteristics. (q) The heatmap shows the predicted relative contribution of lung cell types to the association of protein biomarker signatures in plasma with lung function (forced vital capacity - FVC). Patients were split in two groups, one with a mild decline in lung function [FVC 60-100%] and one with severe loss of lung function [FVC 20-60%] and compared to healthy age matched controls.

The most strongly anti-correlated transcriptional regulator to the *CRTAC1* associated gene programs in AT2 cells was the transcription factor *SOX4* (Fig. 8h). *SOX4* is regulated by various pathways, including TGF-beta signalling and we recently described *Sox4* together with *Nupr1* as a candidate transcriptional regulator of AT2 cell differentiation upon lung injury in the mouse³⁸. Interestingly, the IPA upstream regulator analysis predicted high activity of *NUPR1* also in AT2 cells with low expression of *CRTAC1* (Fig. 8i). *NUPR1* plays a role in cell stress responses, including DNA damage repair and regulation of the cellular senescence program. The co-regulation of *SOX4* and *NUPR1* expression and activity in both mouse and human AT2 cells in health and disease, suggests that this program may have important functions in AT2 cell de-differentiation. We show that *CRTAC1* expression is strongly anti-correlated to this de-differentiation program.

We modeled the de-differentiation of human AT2 cells in ILD by deriving a pseudotime trajectory using all SFTPC+ AT2 cells and the aberrant basaloid cells across all three cohorts (Fig. 8j). The pseudotime trajectory showed a gradual increase of aberrant basaloid cell markers starting already in still SFTPC+ AT2 cells and peaking in then SFTPC-/SOX4+ aberrant basaloid cells, which also expressed genes such as *SFN* and *TNC* that we had found to be increased in the BALF, and the cytoskeletal protein Cornifin-alpha (*SPRR1A*) (Fig. 8k). Importantly, the downregulation of *CRTAC1* occurs in an early stage of this differentiation process. Even though *CRTAC1* expression in AT2 is specific to humans and not observed in mice, we find that the differentiation trajectory we modeled in human IPF is highly similar to a differentiation trajectory observed in mice after bleomycin injury (Fig. 8l). The pseudotime trajectory of mouse AT2 cells towards the *Krt8+* alveolar differentiation intermediate (ADI) cell state³⁸, shows a similar upregulation of *Sox4*, *Sfn*, *Krt8*, *Fn1*, *Tnc*, and *Sprr1a* as observed in the human trajectory (Fig. 8m). This strongly suggests that aberrant basaloid cells in IPF may be generated from AT2 in an attempt for regenerative repair. Indeed, we find

KRT8+/SPRR1A+/SFTPC- cells in close proximity to SFTPC+ cells in areas of limited fibrotic remodeling, which may represent early stage disease (Fig. 8n).

Clinical decisions are often based on blood biomarker analysis⁵¹. To extend our analysis to the plasma proteome we made use of a recently established high throughput plasma proteomics workflow^{50–52} and generated plasma proteomes from two independent cohorts of ILD patients (Munich, n=30 and Hannover, n=81; healthy age matched controls, n=30; see Table S10 for clinical characteristics and Table S11 for plasma protein quantification). We were able to robustly detect CRTAC1 by mass spectrometry in >80% of the plasma samples (Fig. 8o). The Hannover cohort included more patients with better lung function on average, with samples taken mainly at time of initial diagnosis, while the Munich cohort contained patients closer to end stage disease. Thus, by construction, we did not expect a perfect match of these two cohorts. Upon correlation with forced vital capacity (FVC %) we identified a shared panel of proteins in both cohorts that were either positively or negatively associated with the lung function outcome (Fig. 8p). As expected, CRTAC1 showed a positive correlation with lung function in both patient cohorts.

Finally, we investigated the relative contribution of cell types/states to the protein biomarker signatures in plasma that correlated with lung function (Fig. 8q). We divided the patients into two groups representing mild and severe disease based on lung function (FVC %) and compared these two groups with healthy controls. We observed a gradual increase of proteins potentially derived from lung fibroblast subsets and plasma cells as well as a gradual reduction of proteins potentially derived from lung endothelial cells, alveolar macrophages, AT2 cells, and mDC2 (Fig. 8q).

In conclusion, our data and analysis suggest that plasma proteomes harbor protein biomarker signatures that report the status of cell states in health and disease. Here, we demonstrated that the AT2 derived CRTAC1 protein in ILD patient plasma and BAL fluid correlates with lung function and reports the loss of AT2 cell identity during disease progression.

Figure S7. Cell-type specificity of CFHR1, SSTR2 and CRTAC1. (a, b) The dotplots illustrate cell type specific mRNA expression patterns for CFHR1, SSTR2 and pericyte markers (a) as well as CRTAC1 and the AT-2 cell marker SFTPC (b).

6. The paper should include histological validation (IHC/IF) of protein expression in relation to the scRNAseq results. The leap from scRNAseq to secreted protein is very large, and requires additional analyses of the proteins in the tissue context, with respect to the cell types identified by scRNAseq.

To include validation of protein expression in relation to the scRNAseq results, we confirmed specific examples with immunofluorescence (IF) and immunohistochemistry (IHC). We (1) confirmed the disease specific upregulation of the 14-33 protein sigma (SFN) in several epithelial cell populations in situ. Furthermore, we could show that those cells are able to produce ECM proteins involved in tissue repair and also regulated in the BALF proteomes with lung function, such as TNC (Figure 6). We also (2) discovered a disease specific pericyte state, which we localized to small vessels using IF and IHC analysis, correlating the disease associated pericyte state with Ashcroft scores for fibrotic remodelling in tissue sections from >50 ILD patients (Figure 7). Finally, we (3) modeled the differentiation trajectory of AT-2 cells towards ILD specific aberrant basaloid cells. The trajectory was characterized by CRTAC1 levels in BALF and plasma (Figure 8). Of note, we tested all commercially available antibodies for CRTAC1, which unfortunately produced either no or totally unspecific staining, even reproducing the unclear images of the Human Protein Atlas webpage. Therefore, we stained for highly correlated marker genes SPRR1A, KRT8 and SFTPC as shown in Figure 8.

Results section page 12:

Stratification along the protein and cell type specific t-values revealed the expected inverse correlations between protein association with meta lung function and upregulation of the corresponding gene in ILD for several cell types, including KRT5+ basal cells and alveolar epithelial cells (Fig. 6a-f). To histologically validate some of the most significantly regulated proteins in BALF and put these into the context of the cell types identified by scRNAseq, we performed immunofluorescence analysis (Fig. 6g-j). Increased expression of the extracellular matrix protein Tenascin-C, which is a known marker in tissue repair and was also upregulated in mouse lung injury (Fig. S6f), was found in IPF tissue sections in KRT5+ basal cells in the 'bronchiolized' distal lung (Fig. 6g). As predicted by the scRNAseq analysis these cells also co-expressed increased levels of the 14-3-3 protein sigma (SFN), which is an adaptor protein with possible functions in epithelial cell growth and regulation of the p53 pathway³⁷ (Fig. 6h, i). SFN has recently also been described as a marker in a novel transitional stem cell state that transiently appears during lung regeneration^{30,38}. This transitional stem cell state is highly similar to the aberrant basaloid state discovered in IPF^{28,29}, which also expresses increased amounts of SFN. The transitional stem cell state in lung injury repair is also characterized by increased levels of KRT8 expression³⁸, and interestingly, we found that cells with increased levels of KRT8 in metaplastic epithelial areas in IPF lung also co-expressed increased levels of SFN compared to controls (Fig. 6j). Thus, in summary, our in situ validation suggests that the observed negative correlation of TNC and SFN protein in BALF is the consequence of a general upregulation of these markers in several epithelial states within the metaplastic and 'bronchiolized' epithelium in IPF.

Figure 6. Cell type specific transcriptional ILD signatures translate into protein signatures associated with lung function in the BALF. (a, b) Scatter plots stratify genes based on the protein lung function associations (y-axis) and cell type specific ILD associations (x-axis). The size of the dots represents the detection level of each gene in the corresponding cell type. Colors highlight genes with marginal associations at the protein and RNA levels. (c, d) The box plots illustrate differences in mRNA detection for the indicated genes between tissues from end stage lung fibrosis patients in (c) basal cells, and (d) alveolar epithelial cells when compared to controls. (e, f) The scatter plots show the positive correlation of the indicated proteins in BALF (MS-intensity, x-axis) with meta lung function (y-axis). (g-j) Immunofluorescence analysis of the indicated proteins and cell type markers in IPF (n=3) and control samples (n=3).

Results section page 14:

‘The disease specific expression of SSTR2 in PDGFRB+ pericytes was confirmed using immunofluorescence microscopy (Fig. 7f) and immunohistochemistry (Fig. 7g). SSTR2+/PDGFRB+/DES- pericytes were found around remodeled vessels that had a thickened layer of DES+/PDGFRB+ smooth muscle cells in ILD. PDGFRB+/DES- pericytes were negative for SSTR2 in control lungs with normal thickness of the smooth muscle cell layer (Fig. 7f). We quantified the SSTR2 immunohistochemistry signal in 79 tissue sections from 53 ILD patients and 26 control patients, and correlated this signal with the severity of fibrotic remodeling using an Ashcroft scoring (Fig. 7h). The SSTR2 levels were strongly associated with high Ashcroft scores, indicating that the SSTR2+/CFHR1+ pericyte state is correlated with the severity of fibrosis.’

Figure 7. An SSTR2+/CFHR1+ pericyte state correlates with progression of fibrotic remodeling.(f) Immunofluorescence analysis of SSTR2, the pericyte cell type marker PDGFRβ and a marker for smooth muscle cells DESMIN in IPF (n=3) and control samples (n=3). (g) Representative images of immunohistochemistry analysis of SSTR2 protein expression in tissue regions (n = 474) from ILD patients (n = 53) and control patients (n = 26). (h) Correlation of immunohistochemistry signal of SSTR2 with Ashcroft scores.

Results section page 18:

“The pseudotime trajectory of mouse AT2 cells towards the Krt8+ alveolar differentiation intermediate (ADI) cell state³⁶, shows a similar upregulation of *Sox4*, *Sfn*, *Krt8*, *Ftn1*, *Tnc*, and *Sprr1a* as observed in the human trajectory (Fig. 8m). This strongly suggests that aberrant basaloid cells in IPF may be generated from AT2 in an attempt for regenerative repair. Indeed, we find KRT8+/SPRR1A+/SFTPC- cells in close proximity to SFTPC+ cells in areas of limited fibrotic remodeling, which may represent early stage disease (Fig. 8n).”

Figure 8. CRTAC1 protein abundance in BALF and plasma proteomes reports AT2 cell health. ... (n) Immunofluorescence analysis of SPRR1A, KRT8 as well as SFTPC in IPF (n=3) and control samples (n=2).

Minor comments:

1. The authors should add a brief mention of the scRNA process in the results. It is not mentioned at all.

We now mention that we used Dropseq to generate our scRNAseq data:

“Dimension reduction was used to visualize a data manifold representing the gene expression space of 41,888 single cells, generated by using the Dropseq workflow (Macosko et al. 2015).”

2. Figure 1a-b: The authors indicate that cell populations found mainly in patients were called 'activated'; however, AM activated doesn't seem to be in the patients, but higher levels are seen in the controls.

We added a paragraph explaining what the activated AMs are in results section page 6. The frequency of these cells is clearly increased in disease as shown in our Figure S2 (also shown above).

“In macrophages (Fig. 2h), the normal alveolar macrophage phenotype that is marked by high FABP4 expression, is replaced by an ILD associated cell state that features high expression of *SPP1* (Osteopontin)¹⁸, which we termed activated AM. These cells also express higher levels of the CCR2 ligand CCL7, which is a chemoattractant potentially involved in the recruitment of fibrocytes and profibrotic macrophages³⁴⁻³⁶.”

For the second cell state we had termed 'activated' (“pericytes activated”), we performed additional analysis and added a new figure to the revised manuscript. The analysis can be found in Figure 7 of the revised manuscript.

3. I suggest to change the color code of figure S1b. A clearer separation of the colors of the controls vs. the disease would help interpreting the results (rather than comparing to the main figures).

We have generated 2 panels instead - one showing all donors in separate colors and the ILD samples in grey, and the second panel showing all 3 ILD patients in separate colors and all donors in grey.

Figure S1. Clustering analysis and cell type annotation reveals 45 distinct cell type identities in human lung parenchyma. ... (b,c) UMAP embedding displays identified cell types, colored by individual control patients (b) and IPF patients (c).

4. *The integration of scRNAseq of three cohorts mentions 63 patients, however the specific numbers per cohort (patient/control) don't add up. Should be revised.*

We thank the reviewer for recognizing this mistake. We corrected the numbers accordingly:

Total: 32 ILD + 29 controls (=61); Chicago: ILD n=9, controls n=8 (= 17); Nashville cohort: ILD n=20, controls n=10 (= 30); Munich cohort: ILD n=3, controls, n=11 (=14).

5. *The sentence "Gene expression changes in disease were more similar within the epithelial, mesenchymal and leukocyte lineages (Fig. 2d)." (p4) is not clear. What do the authors mean by changes being more similar? Should be revised.*

The up and downregulated genes in fibroblasts are more likely to be also regulated in other mesenchymal cells as in leukocytes and vice versa. We changed the sentence to the following:

“Gene expression changes in disease were most similar in cell types within the respective epithelial, mesenchymal and leukocyte lineages (Fig. 2d). This means that for instance the up and downregulated genes in fibroblasts are more likely to be also regulated in other mesenchymal cells as in leukocytes and vice versa.”

6. *Integration with bulk RNAseq should be more detailed. How many patients were analyzed? What are the biological insights from this integration?*

We added the number of patients to the description of results. Bulk integration allowed us to several additional conclusions: (1) we were able to identify cell state shifts that most likely precede advanced bronchiolization in IPF disease progression (Figure 3); (2) we were able to disentangle gene expression and cell type frequency shifts (Figure 3); (3) most importantly,

we were able to show that the random forest trained with BALF proteome features correctly predicts a gradual decline of lung function across the bulk RNA-seq data from micro-CT staged IPF tissue samples (Figure 5).

7. Reference to figure 2e does not seem to be in the right place.

We corrected the reference to Figure 2e.

8. The distinction between ELF and tissue leakage proteins is not clear. I presume the authors mean that ELF proteins (as they defined them) are expected to have specific functions in the fluid, as opposed to randomly secreted proteins. However, epithelial lining fluid also includes the tissue leakage proteins and it is not known whether they may also have extracellular function in the fluid. Their selection of proteins to focus on should be better explained.

Our analysis in FigS3 provides different levels of information: (1) which proteins are enriched in BALF relative to total tissue, (2) which proteins in BALF are also detected in plasma, (3) which proteins are enriched in BALF relative to plasma. This information is relevant to judge if a protein is locally produced and if it is likely actively secreted into the lumen. We do not claim this implies specific functions or refutes a potential function of leaked proteins within the ELF.

9. Figure S3d should indicate what the color scale refers to.

The dotplot was moved to Figure S3f and the information was added to the figure in the revised manuscript. The color scale of the dotplot refers to the normalized gene expression, while the dot size indicates the percentage of cells expressing the respective gene.

10. Method section of the proteomic analysis indicates a non-linear gradient of water-acetonitrile. The details of this gradient should be indicated especially since these are non-standard running conditions. Normally, gradients do not reach 60% buffer B.

We have now clearly specified in the Methods sections of the revised manuscript:

“Gradient: 5-30%B, for 150min; 30-40%B, for 45min; 40-60%B, for 15 min; 60-95%B, for 10min, 95%B, for 10min; 95-5%B, for 5min“

11. What are the details of the trypsin digestion? Usually runs are longer than 1h (e.g. ON). Which trypsin was used here?

Details of the proteomic digestion were added to the Methods section of the revised manuscript:

“The digestion was performed using LysC (Wako, #129-02541, 10 AU, in ABC) and Trypsin (Trypsin from porcine pancreas: Sigma-Aldrich, T6567) at a 1:50 enzyme to protein ratio in 600 mM Gdm, 800 mM urea, and 3% acetonitrile at 37°C, shaking, overnight.”

12. *Biognosys section of DIA data analysis should move to the next section in the methods rather than the MS part.*

As suggested by the reviewer we adjusted the Methods section.

Referee #3 (Comments on Novelty/Model System for Author):

The data presented in this manuscript are well-analyzed and the overall utility is high. Some of the transcriptomic data are from published datasets, limiting the novelty, but the new analysis and addition of an orthogonal proteomic dataset enriches these datasets. The authors could add some detail clarifying what was learned from combining multiple transcriptomic datasets on beyond what is already known. The potential for medical impact is high as this is a devastating and poorly understood disease, but it would be nice to have further clarity as to how this approach would be for identifying subclinical biomarkers prior to lung function decline.

Referee #3 (Remarks for Author):

In this manuscript, Mayr/Simon et al. describe an integrated computational analysis combining single cell transcriptomic and bulk proteomic measures taken from donors with pulmonary fibrosis and matched controls. By combining their own datasets with previously published analyses, they generate a large single cell dataset of 63 pulmonary fibrosis patients. In parallel, they measure biomarkers in lung lavage and plasma from a separate patient cohort. By developing new computational methods to analyze these datasets in tandem, they identify the type 2 cell protein CRTAC1 as a biomarker of lung function decline in pulmonary fibrosis.

Overall, this effort represents an important compilation and new analysis of novel and publicly available datasets that will serve as a valuable resource for the pulmonary fibrosis field. The motivation for the study is clear and obviously important. In particular, the idea of using proteomic biomarkers to predict cell state changes in difficult-to-access tissues is exciting, and this approach is a potential proof-of-principle for this concept. With this in mind, some aspects of the current analysis make the interpretation difficult to follow, and the full implications of this work are not entirely clear. In particular, a major challenge in presenting large datasets is in clearly outlining the rationale behind the selected analyses

and how the authors reached a given conclusion. While it is of course impossible to follow the data down every possible path, it is important for the reader to understand the rationale behind each decision and interpretation and for the analysis to be well-grounded in prior efforts. Addressing the major points below may help the authors to clarify their findings and the potential impact for the field:

We thank the reviewer for the positive evaluation of our work. Please find our point by point reply below.

Major Points

1. Cell type assignments and identification of disease-associated changes

The authors present an in-depth analysis of single cell data gathered across several patient cohorts from three sites. Overall, the integration of these datasets is well done, and in particular the comparisons of changes in from healthy to disease states across these datasets is useful for identifying genes important to the pathogenesis of IPF. However, throughout the manuscript it is difficult to keep track of the changes in cellular identity occurring with disease onset and progression. Further clarification of the results might illuminate their interpretations and conclusions more fully. In particular, they seem to not always distinguish between changes in the number of a certain cell type (for example, type 2 cells are known to be lost in pulmonary fibrosis) or in the identity of a certain cell type (are there genes expressed in IPF type 2 cells that are not expressed in healthy controls?). Throughout the manuscript, this makes it difficult to follow their interpretations and to ground this new analysis with what is already known from prior studies about the etiology of this disease.

Disentangling effects of disease on cell frequency versus effects on gene expression on cell type level (cell state changes) is indeed very important and now analyzed in detail for the revised manuscript. Single cell transcriptomes clearly enable us to determine changes of gene expression on cell type level (e.g. CRTAC1 is downregulated in SFTPC+ AT2 cells in ILD). However, changes in cell type frequency can in addition affect total abundance of biomarkers in the proximal lavage fluid. Thus, in the case of CRTAC1 the downregulation of gene expression and the reduced number of AT2 in ILD will have an additive effect on protein abundance in the lavage fluid and plasma. In order to systematically annotate regulated proteins with this type of information we performed additional analysis shown in Figure 5f of the revised manuscript.

In this analysis we asked (1) if a regulated protein is expressed in a cell type that shows a significant frequency in bulk deconvolution of donor controls and ILD patient samples, if (2) the protein is also regulated on gene expression level, or (3) both. We then visualized if the gene and protein level changes agreed, thus giving a better clue how discrepancies may arise from opposing phenomena (e.g. downregulated gene expression but increased cell type frequency or vice versa).

Results section page 12:

Next, we aimed to understand the relationship between cell type specific transcriptional changes and distinct signatures in the bronchoalveolar lavage fluid. To accomplish this aim, we integrated the results from two multivariate regression analyses: 1) protein associations with meta lung function and 2) cell type specific RNA associations with ILD status. Correlation of the t-values derived from both regressions revealed that the greatest correspondence between protein signatures associated with meta lung function and expression changes occurred in club and basal cells as well as the alveolar epithelium (Fig. 5e). However, the measured bulk protein profiles in BALF can be affected by two types of alterations: 1) changes in cell type frequency and 2) changes in gene expression. Therefore, we inferred cell type frequency changes by performing deconvolution analysis on a large

bulk expression data set containing ILD patients and healthy controls from the Lung Tissue Research Consortium (LTRC; GSE47460). The top regulated BALF proteins were often altered both on gene expression and cell type frequency levels (Fig. 5f), which importantly can be distinguished using scRNASeq data.

Figure 5... (f) The heatmap illustrates gene expression changes associated with lung fibrosis across indicated cell types (columns) for selected BALF protein biomarkers (rows). The dotplot visualizes the frequency changes of the indicated cell types inferred from the deconvolution of bulk mRNA data of ILD samples compared to controls (GSE47460). Samples used in this study from the LTRC n= 254 ILD patients and n= 108 controls.

- A major advance in this paper is the careful integration of multiple datasets from multiple studies. As there are many published datasets on both healthy and IPF human lung (including the papers describing the datasets used in this paper), it would be helpful for the authors to ground their analysis more substantially in this prior work. In particular, do they learn anything new from this analysis that was not observed in these prior studies?

Integration of three datasets increased the power of statistical analysis and enabled us to derive robust gene expression changes reproducible across cohorts. In contrast to prior work we provide tables that report these robust disease associated gene expression changes.

Results section page 5:

Despite small differences in sequencing depth, differentially expressed genes showed very good agreement between the three independent patient cohorts (Fig. 2e-h). The upregulated *KRT17* gene (Fig. 2f) in alveolar epithelial cells was recently defined as a marker for the novel aberrant basaloid cells in IPF^{28,29}. These fibrosis specific cells feature a cellular senescence signature³⁰, including high expression of *CDKN2A* (encoding for p16; Fig. 2f). We also corroborate previous studies by showing that in fibroblasts (Fig. 2g) the expression levels of *DIO2*, encoding for the thyroid hormone activating enzyme iodothyronine deiodinase³¹, and the circulating *CXCL14*³²⁻³³ chemokine are increased. In macrophages (Fig. 2h), the normal alveolar macrophage phenotype that is marked by high *FABP4* expression, is replaced by an ILD associated cell state that features high expression of *SPP1* (Osteopontin)¹⁸, which we termed activated AM. These cells also express higher levels of the *CCR2* ligand *CCL7*, which is a chemoattractant potentially involved in the recruitment of fibrocytes and profibrotic macrophages³⁴⁻³⁶. These examples manifest that integration of scRNAseq datasets facilitates highly powered and robust differential gene expression analysis, which represents a valuable resource to the research community.

The power of a multi-cohort design is immediately aiding interpretation of observed phenomena. For instance the *SSTR2+* pericyte state was never described before and was not identified in all individuals because of low cell numbers. Using an integrated data object it became however quickly clear that this cell state occurs specifically in disease in several patients across cohorts. This novel *SSTR2+* pericyte state is described in detail in Figure 7 of the revised manuscript. We associate this cell state with the degree of fibrotic remodeling (Ashcroft scores), identify putative transcriptional regulators and show that the complement regulatory factor *CFHR1* is expressed highly specifically in these cells and correlates with lung function decline in the lavage fluid.

In the revised manuscript we also analyzed cell type frequency changes across cohorts and performed deconvolution analysis that highlights cell frequency changes along disease progression. We observed the appearance of the novel aberrant basaloid cell state described in earlier work used for the integration already in early stage IPF-1. Importantly, the pseudotime modeling of AT2 cell transition towards this basaloid cell state (Figure 8) highlights that the *CRTAC1* biomarker discovered in this study marks an early stage of AT2 cell dedifferentiation towards basaloid cells, which is a clear advance over prior studies. We also added novel in depth analysis of gene-gene covariation across single cells in the revised manuscript, which revealed gene programs associated with health and disease in AT2 cells (Figure 8). Our analysis goes beyond previous work and describes gene signatures associated with early AT2 cell dedifferentiation.

• In Figure 1, they choose to subset the data based on one marker gene per population, but do not comment on whether this marker gene changes in disease. For example, they choose to look at EPCAM+ epithelial cells, but EMT (and associated loss of EPCAM) in epithelia is associated with pulmonary fibrosis - is it possible they are missing some unique epithelial populations (or intermediate "EMT" populations) in patient samples using this approach? Are there cells that don't fall into these categories?

The four subsets in Fig. 1 cover all clusters in the data. The reviewer refers to possible EMT in IPF - we have looked at this in detail in our recently published manuscript on a novel transitional stem cell state (Krt8+ ADI) appearing in mouse lung regeneration. The KRT5-KRT17+ aberrant basaloid cells, which are very similar to Krt8+ ADI described in mouse, show several features of EMT such as upregulation of FN1 and COL1A2. These cells are however still EPCAM+ and not anywhere near the transcriptional identity of fibroblasts. Using the single cell analysis we currently conclude that full EMT does not occur in mouse and human lung fibrosis.

• They mention the assignment of "activated cell states" that are associated with IPF, but nowhere in the manuscript are these states enumerated and characterized. It seems like the analysis they performed should include all of these data; it would be nice to have them included in the manuscript.

We added a paragraph explaining what the activated AMs are in results section page 6. The frequency of these cells is clearly increased in disease as shown in our Figure S2 (also shown above).

"In macrophages (Fig. 2h), the normal alveolar macrophage phenotype that is marked by high FABP4 expression, is replaced by an ILD associated cell state that features high expression of *SPP1* (Osteopontin)¹⁸, which we termed activated AM. These cells also express higher levels of the CCR2 ligand CCL7, which is a chemoattractant potentially involved in the recruitment of fibrocytes and profibrotic macrophages³⁴⁻³⁶."

For the second cell state we had termed 'activated' ("pericytes activated"), we performed additional analysis and added a new figure to the revised manuscript. The analysis can be found in Figure 7 of the revised manuscript.

• Related to this, in some analyses it is difficult to understand whether they are measuring changes in cell identity/gene expression or the number of a certain cell type. For example, is Figure 1j showing that ciliated cells are increased in later stage pulmonary fibrosis, or that there are certain genes increased within the ciliated cell population in pulmonary fibrosis. If it is the former, what are the genes that are changing in this population?

We added additional analysis contrasting inferred changes in cell type frequency with changes in gene expression to the revised manuscript. We have ranked the cell types based on their contribution to the BALF proteome signatures (Fig. 5e). The protein signature is a

composite representation of both cell type frequency and gene expression changes. Single cell analysis enables us to disentangle both layers of regulation.

Results section page 12:

Next, we aimed to understand the relationship between cell type specific transcriptional changes and distinct signatures in the bronchoalveolar lavage fluid. To accomplish this aim, we integrated the results from two multivariate regression analyses: 1) protein associations with meta lung function and 2) cell type specific RNA associations with ILD status. Correlation of the t-values derived from both regressions revealed that the greatest correspondence between protein signatures associated with meta lung function and expression changes occurred in club and basal cells as well as the alveolar epithelium (Fig. 5e). However, the measured bulk protein profiles in BALF can be affected by two types of alterations: 1) changes in cell type frequency and 2) changes in gene expression. Therefore, we inferred cell type frequency changes by performing deconvolution analysis on a large bulk expression data set containing ILD patients and healthy controls from the Lung Tissue Research Consortium (LTRC; GSE47460). The top regulated BALF proteins were often altered both on gene expression and cell type frequency levels (Fig. 5f), which importantly can be distinguished using

scRNASeq data.

Figure 5... (f) The heatmap illustrates gene expression changes associated with lung fibrosis across indicated cell types (columns) for selected BALF protein biomarkers (rows). The dotplot visualizes the frequency changes of the indicated cell types inferred from the deconvolution of bulk mRNA data of ILD samples compared to controls (GSE47460). Samples used in this study from the LTRC n= 254 ILD patients and n= 108 controls.

- In Figure 2., the use of available data to track disease progression and map samples to early disease states is nice, but it isn't clear what the actual interpretation of this analysis is.

Is the myofibroblast signature upregulated in the early stages of disease because fibrotic infiltration (and an increase in fibroblast gene expression in the tissue) is an early event, or because of early transformation of existing fibroblasts towards a more disease-like myofibroblast state? After clarifying which of these hypotheses they intend to propose, can they provide further evidence (even in the form of a clearer analysis of this data) to support this claim?

In this analysis we infer relative cell type frequencies based on the most significant marker genes defining the individual cell populations. As shown in Figure 3e/f the frequency of myofibroblasts is predicted to be already high in IPF stage-1. Indeed, in the bulk sample this can be driven by both the necessary change in gene expression in pre-existing fibroblasts, as well as the increase in relative frequency of myofibroblasts. We analyzed how many of the myofibroblast genes shown in panel f of Figure 3 are indeed also significantly upregulated in fibroblasts in the single cell analysis (see below). This analysis shows that most of these genes are indeed significantly upregulated in myofibroblasts compared to all other fibroblasts, indicating that this cell state transition happens early in IPF stage-1.

- Overall, it would really clarify their findings if they were able to more clearly relate the observed data to changes in cell number and identity in pulmonary fibrosis, in particular observations known from prior work (for example, do they see evidence of bronchiolization, infiltration/transformation of macrophages, loss of type 2 cells) and from the analysis of the datasets they use in this paper from prior single cell studies. Again, this is likely just a matter of including the relevant graphs in the supplemental data or making an interactive data browser available.

As explained above we have done new analysis to better relate fluid proteomes with cell frequency as well as cell state changes. Our analysis confirms known phenomena consistent with prior work such as bronchiolization (described in several sections of the revised manuscript), altered macrophage states that appear in higher frequency in disease, as well as the loss of type-1 and type-2 cells (see Figure 3). All of these changes are described and reflected across data modalities as outlined.

Figure 3. Disease progression alters cell type frequencies. (a) Principal components one (x-axis) and two (y-axis) illustrate the cell type frequencies in reduced dimensions. The shape of each point corresponds to the study cohort and the points are colored by disease phenotype. (b) Boxplots display principal component two across ILD patients (orange) and control samples (purple) for the three study cohorts. (c) Boxplot depicts random forest model prediction accuracies derived from five fold cross-validation using the original and permuted labels. (d) Barplot shows the random forest importance scores for the top ten most informative features. (e) The heatmap shows changes of the indicated cell type signatures in published bulk RNA-seq data (GEO GSE124685) across different histopathological stages that represent increasing extent of fibrosis from stage 1-3, as determined by quantitative micro-CT imaging and tissue histology²². **Samples used in this study were 10 IPF and 6 control patients.** (f) The heatmaps show z-scores for the individual marker genes of the indicated cell types across IPF stages and controls.

2. Analysis of proteomic datasets and comparison to single cell sequencing results

The authors present a new dataset of proteomic analysis of lavage fluid and plasma from patients with pulmonary fibrosis. From these data, they validate known biomarkers and propose new signatures correlated with clinical features and disease progression. Finally, they map between this dataset and their single cell transcriptomic analysis to test the utility of lavage/plasma biomarkers for predicting cell state changes in disease-affected tissue. While biomarker studies have been done in IPF, this is a large dataset and represents a first effort to compare with sequencing dataset. The data and analysis approach are of high-quality and of clear use to the field. However, further clarification on a few points would help support the author's interpretation of these result. In particular, as in Major Point (1), it is occasionally difficult to follow the rationale for the selected analysis approach and the consequences for the subsequent interpretation of these results:

We thank the reviewer for the positive comments.

• *The authors choose to focus their analysis on secreted proteins and limit their interpretation of proteins released as a likely consequence of tissue damage. It is not clear why they make this decision, as protein accumulation in the lung lining fluid and in plasma is a clear marker of tissue damage associated with disease. Can the authors clarify the motivation behind this decision, and how this might change their interpretation of the results?*

This is a misunderstanding. We did not focus the analysis on only secreted proteins. We only made additional efforts to assign categorical annotations to BALF protein (e.g. enriched over tissue, or enriched over plasma). For correlation with clinical parameters such as lung function we used all proteins irrespective of their proposed cell and tissue compartments.

In the revised manuscript we quantitatively compared BALF and plasma proteomes. Many proteins are detected both in plasma and BALF - within these proteins we find proteins that are having a higher abundance in the BALF (Fig S3b). These proteins, including CRTAC1 and known cell type markers such as BPIFB1 expressed by airway secretory cells and AT2 specific SFTPB, are thus produced locally and transpire to the plasma.

Results section page 8:

‘From 124 patients (95 ILD and 29 non-ILD) that passed quality control criteria, we quantified a median number of 835 proteins per individual patient, resulting in a total of 1513 unique proteins that were quantified in at least 20 patients (Fig. 4b and Table S6). This is a very good depth of analysis given that BALF is difficult to analyze by mass spectrometry because of the high dynamic range of protein copy numbers **present due to plasma protein leakage. To illustrate a correlation of the extent of plasma protein leakage with the depth of proteome analysis in our BALF samples, we quantified the mass fraction of the top 100 abundant proteins in patient plasma in the BALF proteomes. The number of detected proteins in BALF inversely correlated with the relative proportion of plasma proteins present (Fig. S3a), however this was not strongly associated with plasma LDH, which serves as a marker for tissue damage in patient blood. Using a quantitative comparison of proteins detected in both the ILD patient plasma and BALF proteomes revealed proteins that are detected in plasma proteomes but are quantitatively enriched in BALF, suggesting they are produced locally in the lung and then transpire to the plasma (Fig. S3b).**’

Figure S3. Defining the epithelial lining fluid proteome in a large interstitial lung disease patient cohort. (a) The fraction of plasma proteins (defined by top 100 abundance rank in plasma) is shown as a percentage of total mass fraction of all identified proteins in BALF. Note that this fraction negatively correlates with the total number of identified proteins in BALF. The colors represent plasma LDH values of individual patients. (b) The scatter plot shows mean MS-intensities of proteins identified in both plasma and BALF from all ILD patients. The red circle indicates locally enriched proteins that are more abundant in the lung compared to plasma samples.

- *Much of the data validates known features of pulmonary fibrosis, such as an increase in immune infiltration in the lung associated with measurement of cytokine and other pro-inflammatory proteins. It is not clear from the presentation of these data what exactly the proteomic data are showing beyond this finding. For example, are the authors able to distinguish increased immune cell infiltration versus a change in protein composition or cytokine secretion from these cells and therefore make a prediction about potential mechanistic drivers of disease?*

We described associations of immune cell counts in BAL with protein profiles in the correlation analysis with available metadata (Figure 5). All these associations are available in the supplementary data tables.

- *The analysis of CRTAC1 as a biomarker is nicely done and it is clear that this gene is downregulated in IPF (and consequently the loss of this protein can be readily measured as a biomarker). It seems that CRTAC1 is downregulated in type II cells in disease and that type II cells are lost in disease progression. It seems clear that there should be an association between loss of type II cells and loss of lung function. Do they see a similar response in other genes/proteins associated with type II cells, or is there something unique about this gene beyond its expression in this cell type? If the latter, can the authors speculate as to the importance of this gene in disease?*

We have substantially extended our analysis on CRTAC1, which is now presented in Figure 8 of the revised manuscript. CRTAC1 shows a strong and specific association with AT2 cells (Fig. S7). To address if the very significant association of CRTAC1 with lung function is merely due to the loss of AT2 cells or even more specific, we performed (1) gene-gene covariation analysis within SFTPC+ AT2 cells as well as (2) pseudotime modeling of the dedifferentiation of AT2 cells towards the aberrant basaloid cells (Figure 8). Loss of CRTAC1 marks an early step of AT2 dedifferentiation. The functional relevance of CRTAC1 as secreted protein in maintenance of alveolar identity is currently unknown and an interesting avenue for future research.

Results section page 16:

Analysis of transcriptional regulators revealed that the expression of the transcriptional activator C/EBP-delta (CEBPD) was highly correlated with CRTAC1 expression (Fig. 8h). CEBPD expression and activity is induced by glucocorticoids and has a role in AT2 cell differentiation during lung development^{45,46}. Interestingly, the levels of CRTAC1 in isolated human AT2 cells increase upon differentiation with glucocorticoids⁴⁷, which are known to be essential for alveolar maturation in lung development⁴⁸. Thus, we speculate that the downregulation of CRTAC1 in AT2 cells of ILD patients may hint at currently uncharacterized changes in glucocorticoid signaling in these cells. We also performed upstream regulator analysis in Ingenuity Pathway Analysis (IPA) to predict the activity of transcriptional regulators based on the correlated or anti-correlated gene profiles (Fig. 8i). This analysis identified ETV5 as a potential regulator of the CRTAC1 correlated gene program in AT2. This is in line with the fact that ETV5 has been shown to be essential for AT2 cell maintenance in vivo, as deletion of *Etv5* from mouse AT2 cells produced gene and protein signatures characteristic of differentiated alveolar type I (AT1) cells⁴⁹. Thus, our finding here is consistent with the notion that CRTAC1 is associated with a normal healthy and highly differentiated AT2 cell state.

• The idea of developing a biomarker to measure lung function decline is exciting, but its clinical utility relies on the ability to detect lung function decline at a sub-clinical level (e.g. in early stages of disease), as it is relatively simple to measure clinically. The analysis of relative contribution of cell types to the biomarker signature seems really useful and promising for subclinical phenotyping. It would be useful to further understand how they might also detect subclinical or more subtle early changes, what the sensitivity of this approach might be, and how these data could be analyzed and interpreted. Can the authors comment further on how this might be accomplished using this approach?

If CRTAC1 alone will have clinical utility is unclear at this point. We see this as an important proof of concept that blood biomarker profiles may one day be used to monitor cell state identities in tissues (for instance in a longitudinal setting in therapy). Sensitivity and specificity of such an assay will depend on analytical methods used and the selection of marker panels that most specifically reflect cell state identities in tissues.

Interestingly, our notion that CRTAC1 levels drops as soon as AT2 cells start to de-differentiate in an acute inflammatory or injury scenario seems to be confirmed in severe COVID-19 cases:

Discussion section page 20:

‘We propose that CRTAC1 protein levels in plasma and lavage fluids specifically report the AT2 cell health status. This novel biomarker is thus a promising candidate for future prospective trials in various settings, including the monitoring the degree of distal lung involvement during virus induced pneumonia as currently seen in the COVID-19 pandemic. Supporting our hypothesis, a recent preprint illustrates that CRTAC1 is downregulated in plasma of hospitalized COVID-19 patients compared to SARS-CoV-2 negative controls⁵⁷.’

• *Related to this, in Figure 2i-j they measure gene expression changes predicted to map to early stages of disease. Can they use this result together with their later machine learning analysis to predict protein biomarker changes that might occur early in disease?*

We thank the reviewer for this constructive suggestion. Training a random forest with the protein biomarkers in BALF that correlated with the meta-lung function parameter enabled us to predict lung function along the micro-CT staged samples that represent a histopathological progression of disease within individual patients. It was intriguing to see that the BALF signature derived from training across patients that are in different stages of their disease progression (e.g. different loss of lung function) nicely predicted a gradual decline of lung function with higher micro-CT based histopathology score (Figure 5i), indicating that these features quantitatively reflect histopathology in the organ.

Results section page 12:

‘Finally, to test if we could successfully transfer information from the proteomics modality into the scRNAseq data modality we applied machine learning. A random forest was trained on the protein quantification data to predict lung function (DLCO) using a set of protein features which 1) showed high correlation with lung function (DLCO) and 2) had the corresponding transcript detected in the scRNAseq data. Next, the trained model was applied to in-silico bulk scRNAseq data with mRNA expression mapped to proteins (Fig. 5g), which then correctly predicted the direction of lung function changes in the three single cell RNA-seq cohorts (Fig. 5h). In addition, we applied an analogous approach to published bulk RNAseq data of IPF samples from different histopathological stages determined by quantitative micro-CT imaging and tissue histology (GEO GSE124685)²² (Fig. 5i). Our model successfully predicted continuous lung function decline along the histopathological IPF disease stages, indicating that the BALF protein biomarker profiles discovered in this study quantitatively reflect cell state changes during disease progression.’

Figure 5. Protein signatures in BALF predict lung function decline and the corresponding cellular changes. (g) The protein features in BALF were used to train a random forest algorithm to predict lung function. The model was tested on transcriptional signatures from single cell RNA-seq data to correctly predict reduced lung function in end stage lung fibrosis when compared to controls. (h, i) Box plots show predicted lung function changes (DLCO%) in the three single cell RNA-seq cohorts (h) and published bulk RNAseq of IPF samples from different histopathological stages (GEO GSE124685) (i)

Minor Points

- *Early in the manuscript, the authors discuss in detail the effort of the Human Cell Atlas project. Why the focus on this particular effort - are all of the datasets used in this study funded by this group?*

We added a paragraph in the Acknowledgements section mentioning that this study is part of the Human Cell Atlas and LifeTime initiatives.

“This publication is part of the Human Cell Atlas (www.humancellatlas.org/publications/) and the LifeTime Initiative (<https://lifetime-fetflagship.eu/>).”

- *The use of bbknn for batch correction seems to work nicely, but this method relies on a relatively one-to-one mapping of states across the selected "batch" variable. As the authors later use a regression analysis to look into demographic covariates, it seems like they don't necessarily expect these datasets to be well-matched. Would it be possible to use the approach used in Fig S2 to also perform batch correction for Fig. 1?*

The reviewer makes an interesting suggestion to perform batch correction in the gene expression space using regression methods instead of deriving an integrated embedding from the principal component space. Prior work has indeed demonstrated the suitability of regression methods for batch correction in scRNAseq data (Buettner et al Nat Meth 2019).

However, we decided to apply a different analysis strategy for the following reasons. All samples were derived from the same tissue which implies a relatively one-to-one mapping of cell types. However, we expected to observe differences in cell states based on demographic differences. Considering sources of variation in scRNAseq gene expression, cell type identity is expected to have a much greater impact compared to demographic covariates. Therefore, we used BBKNN to integrate transcriptionally similar cells from different studies. The

integrated embedding primarily clustered cells by identity. On a more subtle level we expected demographic covariates to contribute to the variation within clusters. We build our differential expression analysis on this rationale. First we grouped cells of the same type, subsequently we performed differential expression analysis across different states of the same type.

• *In the introduction they mention the clinical utility of biomarkers to measure patient heterogeneity in a complex disease like IPF. They later discuss this in the context of acute exacerbations, but it might also be interesting to learn if they see any evidence for different etiologies or subtypes of pulmonary fibrosis in different patients. Are they able to understand anything further about this from these data?*

We did observe clear differences between different diagnostic entities of ILD (Fig. 5a). In the revised manuscript we analyzed cell type frequencies in more detail. Indeed, we observed greater heterogeneity in the disease samples compared to controls (Fig. 3a):

“Differences in cell type frequencies between individual samples can be caused by true biological differences, as well as differences in cell isolation protocols and scRNAseq platforms used. Indeed, we observed large variance in cell population frequencies across cohorts, and disease conditions (Fig. S2c). After performing dimension reduction using PCA, we observed larger variation in principal components one and two among ILD patients compared to control samples, indicating increased heterogeneity in disease (Fig. 3a).”

Figure 3. Disease progression alters cell type frequencies. (a) Principal components one (x-axis) and two (y-axis) illustrate the cell type frequencies in reduced dimensions. The shape of each point corresponds to the study cohort and the points are colored by disease phenotype...

These results suggest possible existence of disease subtypes. However to fully address this interesting question larger sample size and/or clear disease subtype definitions would be necessary. As more and more scRNAseq data sets become available, we plan to perform even larger integration which will provide sufficient power to clearly study disease subtypes in future work.

- *The patient cohorts used for the proteomic versus transcriptomic analysis are non-overlapping. The authors comment that this is a major limitation of this approach. Can they further clarify the differences between these cohorts and what the specific limitations are for the interpretation of the results? Do they expect to see differences in any particular parameters? Do they have paired transcriptomic/proteomic datasets for any individual to serve as validation for their approach?*

Unfortunately, we do not have any individual with paired transcriptomic/proteomic data. This would indeed provide benefit to the analysis. Nonetheless, using transfer learning techniques we successfully demonstrated that information can be transferred across cohorts. For example, in Figure 5g-i of the revised manuscript we show that training a model based on proteomic data in one cohort generates accurate predictions in two independent cohorts without proteomic data (as shown above).

- *The authors see the strongest association in their proteomic dataset with changes in epithelial lineages, possibly driven by their focus on secreted proteins. The epithelium is highly secretory and responsive to damage and environmental changes. Is this the likely reason for this strong association, or are there other possible interpretations?*

As clarified above the proteomic analysis of BALF was not focused on secreted proteins only. Asking which proteins in BALF have a corresponding gene expression change on cell type level (Fig. 5e) revealed that indeed the epithelial cell identities were dominant. We explain this by the fact that these cells have the closest interface to ELF sampled by the lavage. Nevertheless, the analysis also shows that many of the proteins that are regulated with lung function also show gene expression changes in leukocyte populations such as the alveolar macrophage.

7th Dec 2020

Dear Dr. Schiller,

Thank you for the submission of your revised manuscript to EMBO Molecular Medicine. We have now received the enclosed report from the referees who were asked to re-assess it. As you will see the referees are now supportive and I am pleased to inform you that we will be able to accept your manuscript pending the following editorial amendments:

1. In the main manuscript file, please do the following:

- Reduce key word number to 5.
- Remove the red color font.
- The callout for Figure 5D is missing, please fix.
- Please remove Figures from the main manuscript file and create a Figure Legend Section for the main figures located before the references section.
- In Materials and Methods, include a statement that informed consent was obtained from all subjects and that the experiments conformed to the principles set out in the WMA Declaration of Helsinki and the Department of Health and Human Services Belmont Report.

2. Appendix: Please remove supplementary figures and legends from the main manuscript file. Please merge them into a single separate pdf file called "Appendix". Add a Table of Content on the 1st page. Please update all callouts for Figures to Appendix Figure S#.

3. Please update Tables to Datasets and include the legends within the Dataset with a separate sheet labelled Legend. Please update all Table callouts to Dataset EV#.

4. Author contributions: only 21 out of 23 authors are called out in this section. The missing names should be included. The format needs to be changed and authors' initials should be used. Please refer to any of our published papers for an example.

5. Reference format: citations should be listed in alphabetical order. List 10 co-authors of a paper before to add et al. Please also remove web links from the References. More information can be found here: <https://www.embopress.org/page/journal/17574684/authorguide>

6. Data availability: this section should be placed after the Materials & Methods section.

7. The Paper Explained: EMBO Molecular Medicine articles are accompanied by a summary of the articles to emphasize the major findings in the paper and their medical implications for the non-specialist reader. Please provide a draft summary of your article highlighting

- a. the medical issue you are addressing,
- b. the results obtained and
- c. their clinical impact.
- d. This may be edited to ensure that readers understand the significance and context of the research. Please refer to any of our published articles for an example.

8. For More Information: There is space at the end of each article to list relevant web links for further consultation by our readers. Could you identify some relevant ones and provide such information as well? Some examples are patient associations, relevant databases, OMIM/proteins/genes links,

author's websites, etc..

9. Author checklist needs to be completed. Both co-correspondence authors' names should be on the checklist.

10. Funding information is currently missing from our online submission system - Please add the following information when submitting the revised manuscript: This work was supported by the German Center for Lung Research (DZL), the Helmholtz Association, the Max Planck Society, and the German Federal Ministry of Education and Research (BMBF), project Single Cell Genomics Network Germany, and the European Union's Horizon 2020 research and innovation programme under grant agreement No 874656 (discovAIR).

11. Every published paper now includes a 'Synopsis' to further enhance discoverability. Synopses are displayed on the journal webpage and are freely accessible to all readers. They include a short stand first (maximum of 300 characters, including space) as well as 2-5 one sentence bullet points that summarize the paper. Please write the bullet points to summarize the key NEW findings. They should be designed to be complementary to the abstract - i.e. not repeat the same text. We encourage inclusion of key acronyms and quantitative information (maximum of 30 words / bullet point). Please use the passive voice. Please attach these in a separate file or send them by email, we will incorporate them accordingly.

12. Please provide a "synopsis image" or visual abstract (550px width and 400px height, jpeg format) to highlight the paper on our homepage.

13. As part of the EMBO Publications transparent editorial process initiative (see our Editorial at <http://embomolmed.embopress.org/content/2/9/329>), EMBO Molecular Medicine will publish online a Review Process File (RPF) to accompany accepted manuscripts.

a. In the event of acceptance, this file will be published in conjunction with your paper and will include the anonymous referee reports, your point-by-point response and all pertinent correspondence relating to the manuscript. Let us know if you do NOT agree with this.

I look forward to seeing a revised version of your manuscript as soon as possible.

Sincerely,
Jingyi

Jingyi Hou
Editor
EMBO Molecular Medicine

*** Instructions to submit your revised manuscript ***

To submit your manuscript, please follow this link:

Link Not Available

- 1) a .docx formatted version of the manuscript text (including Figure legends and tables)
- 2) Separate figure files*
- 3) supplemental information as Expanded View and/or Appendix. Please carefully check the authors guidelines for formatting Expanded view and Appendix figures and tables at <https://www.embopress.org/page/journal/17574684/authorguide#expandedview>
- 4) a letter INCLUDING the reviewer's reports and your detailed responses to their comments (as Word file).
- 5) The paper explained: EMBO Molecular Medicine articles are accompanied by a summary of the articles to emphasize the major findings in the paper and their medical implications for the non-specialist reader. Please provide a draft summary of your article highlighting
 - the medical issue you are addressing,
 - the results obtained and
 - their clinical impact.This may be edited to ensure that readers understand the significance and context of the research. Please refer to any of our published articles for an example.
- 6) For more information: There is space at the end of each article to list relevant web links for further consultation by our readers. Could you identify some relevant ones and provide such information as well? Some examples are patient associations, relevant databases, OMIM/proteins/genes links, author's websites, etc...
- 7) Author contributions: the contribution of every author must be detailed in a separate section.
- 8) EMBO Molecular Medicine now requires a complete author checklist (<https://www.embopress.org/page/journal/17574684/authorguide>) to be submitted with all revised manuscripts. Please use the checklist as guideline for the sort of information we need WITHIN the manuscript. The checklist should only be filled with page numbers where the information can be found. This is particularly important for animal reporting, antibody dilutions (missing) and exact

values and n that should be indicated instead of a range.

9) Every published paper now includes a 'Synopsis' to further enhance discoverability. Synopses are displayed on the journal webpage and are freely accessible to all readers. They include a short stand first (maximum of 300 characters, including space) as well as 2-5 one sentence bullet points that summarise the paper. Please write the bullet points to summarise the key NEW findings. They should be designed to be complementary to the abstract - i.e. not repeat the same text. We encourage inclusion of key acronyms and quantitative information (maximum of 30 words / bullet point). Please use the passive voice. Please attach these in a separate file or send them by email, we will incorporate them accordingly.

You are also welcome to suggest a striking image or visual abstract to illustrate your article. If you do please provide a jpeg file 550 px-wide x 400-px high.

10) A Conflict of Interest statement should be provided in the main text

11) Please note that we now mandate that all corresponding authors list an ORCID digital identifier. This takes <90 seconds to complete. We encourage all authors to supply an ORCID identifier, which will be linked to their name for unambiguous name identification.

Currently, our records indicate that the ORCID for your account is 0000-0003-1914-3215.

Link Not Available

12) The system will prompt you to fill in your funding and payment information. This will allow Wiley to send you a quote for the article processing charge (APC) in case of acceptance. This quote takes into account any reduction or fee waivers that you may be eligible for. Authors do not need to pay any fees before their manuscript is accepted and transferred to our publisher.

Photos 400-800 DPI

*Additional important information regarding figures and illustrations can be found at <https://bit.ly/EMBOPressFigurePreparationGuideline>

The system will prompt you to fill in your funding and payment information. This will allow Wiley to send you a quote for the article processing charge (APC) in case of acceptance. This quote takes into account any reduction or fee waivers that you may be eligible for. Authors do not need to pay

any fees before their manuscript is accepted and transferred to our publisher.

***** Reviewer's comments *****

Referee #1 (Comments on Novelty/Model System for Author):

I find that the integrated scRNA-seq and proteomic analysis of body fluid is very interesting, and the association between secreted proteins and the cell of origin may also be applicable to other studies. The authors use state of the art technologies and innovative integration approaches.

Referee #1 (Remarks for Author):

The authors addressed all of my comments properly, and added substantial data and analyses that improved the manuscript. I believe that the manuscript now is stronger and more suitable to the professional audience and the computational and proteomics audience. I therefore find the manuscript suitable for publication.

Referee #3 (Comments on Novelty/Model System for Author):

The authors have made a substantial revision including careful analysis and interpretation of their results. With this in mind, they have presented here a detailed manuscript with thorough statistical analysis and interesting findings. These findings are likely to inform our understanding of pulmonary fibrosis.

Referee #3 (Remarks for Author):

The authors have made a substantial revision to their paper that clearly addresses all of my major points and strengthens the conclusions of the work. I have no additional concerns.

The authors performed the requested editorial changes.

19th Jan 2021

Dear Dr. Schiller,

We are pleased to inform you that your manuscript is accepted for publication and is now being sent to our publisher to be included in the next available issue of EMBO Molecular Medicine.

We would like to remind you that as part of the EMBO Publications transparent editorial process initiative, EMBO Molecular Medicine will publish a Review Process File online to accompany accepted manuscripts. If you do NOT want the file to be published or would like to exclude figures, please immediately inform the editorial office via e-mail.

Please read below for additional IMPORTANT information regarding your article, its publication and the production process.

Congratulations on your interesting work,

Jingyi

Jingyi Hou
Editor
EMBO Molecular Medicine

Follow us on Twitter @EmboMolMed
Sign up for eTOCs at embopress.org/alertsfeeds

***** Reviewer's comments *****

YOU MUST COMPLETE ALL CELLS WITH A PINK BACKGROUND ↓
PLEASE NOTE THAT THIS CHECKLIST WILL BE PUBLISHED ALONGSIDE YOUR PAPER

Corresponding Author Name: Herbert B. Schiller, Fabian J. Theis

Manuscript Number: EMM-2020-12871-V2